# REMOVE360: BENCHMARKING RESIDUALS AFTER OBJECT REMOVAL IN 3D GAUSSIAN SPLATTING

## ABSTRACT

Understanding what semantic information persists after object removal is critical for privacy-preserving 3D reconstruction and editable scene representations. In this work, we introduce a novel benchmark and evaluation framework to measure semantic residuals—the unintended semantic traces left behind—after object removal in 3D Gaussian Splatting. We conduct experiments across a diverse set of indoor and outdoor scenes, showing that current methods can preserve semantic information despite the absence of visual geometry. We also release Remove360, a dataset of pre/post-removal RGB images and object-level masks captured in real-world environments. While prior datasets have focused on isolated object instances, Remove360 covers a broader and more complex range of indoor and outdoor scenes, enabling evaluation of object removal in the context of full-scene representations. Given ground truth images of a scene before and after object removal, we assess whether we can truly eliminate semantic presence, and if downstream models can still infer what was removed. Our findings reveal critical limitations in current 3D object removal techniques and underscore the need for more robust solutions capable of handling real-world complexity. Dataset is available at https://huggingface.co/datasets/simkoc/Remove360.

## 1 INTRODUCTION

Trainable scene representations, such as neural radiance fields (NeRFs) (Mildenhall et al., 2020; Barron et al., 2022; Reiser et al., 2021; Müller et al., 2022; Chen et al., 2024a; Kulhanek & Sattler, 2023; Martin-Brualla et al., 2021) or 3D Gaussians (Kerbl et al., 2023; Lin et al., 2024; Yu et al., 2024; Zhang et al., 2024; Kulhanek et al., 2024; Chen et al., 2024c; Wang et al., 2024) enable photorealistic 3D reconstructions from images, and can be easily enriched with semantic features (Kerr et al., 2023; Shi et al., 2024; Ye et al., 2024a; Wu et al., 2024a; Zhou et al., 2024; Hu et al., 2024; Jain et al., 2024). This allows intuitive search via natural language prompts (Peng et al., 2023; Huang et al., 2024; Takmaz et al., 2025; Liang et al., 2024; Koch et al., 2024), e.g., asking 'find the remote control'. Similarly, it enables intuitive editing (Ye et al., 2024a; Zhou et al., 2024; Chen et al., 2024b; Gu et al., 2024; Choi et al., 2024a), e.g., by asking to 'remove the red armchair in the living room'.

The growing availability of learning-based 3D reconstruction software accessible to non-expert users (sca; Tancik et al., 2023; Yu et al., 2022; RealityCapture2023, 2023; pol; Ye et al., 2024b; pos; lum), coupled with intuitive edit operations based on natural language (Radford et al., 2021; Achiam et al., 2023; Schuhmann et al., 2022), opens up exciting possibilities: Using data casually captured by a smart phone (sca; pol; rea; lum), a user can easily create and edit photorealistic 3D models. At the same time, this raises privacy concerns: users may wish to remove private objects, such as photos, documents, or decorations, before sharing reconstructions online[1]. A key question is whether current editing methods truly remove objects, or whether they leave semantic residuals from which one can still infer what was removed.

This paper is dedicated to this privacy aspect of (mask and language-based) editing of trainable scene representations. We focus on removing objects from scenes and investigating whether state-of-the-art

---

[1]As an example, IKEA provides an app that allows users to scan rooms. The captured data is uploaded to IKEA's servers. Hence, IKEA recommends to physically remove private parts before scanning. In contrast, the systems envisioned in this paper allow to perform the removal virtually after the scan, which is more practical.

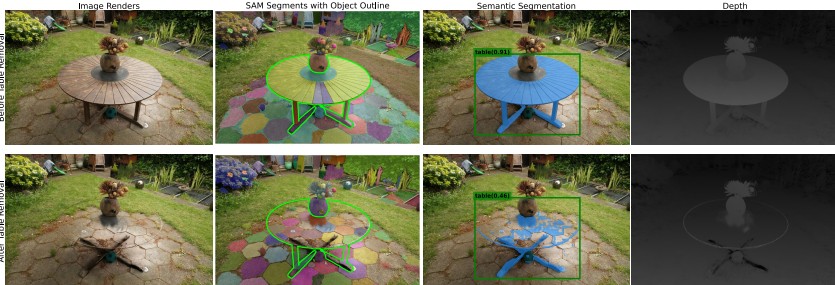

Figure 1: **Detecting (semantic) traces left behind after removing an object from a 3DGS reconstruction.** When there remain residuals of the object, and they can be reasoned over, the object removal is imperfect. We measure the presence of residuals with off-the-shelf semantic models and with depth. Top-Bottom: 3DGS Kerbl et al. (2023) scene before and after table removal. Left-Right: RGB renderings, SAM Kirillov et al. (2023) masks overlay with pseudo-ground-truth object outline, GroundedSAM Kirillov et al. (2023); Liu et al. (2023); Ren et al. (2024) overlay, depth renderings.

removal methods leave residuals that enable us to reason about what content was removed. Contrary to previous works (Cen et al., 2023a; Choi et al., 2024b; Mirzaei et al., 2023) that focus on foreground / background segmentation, we are interested in whether the residuals of the objects remain in the scene after removal and whether the residuals can be reasoned over (see Fig. 1). To the best of our knowledge, we are the first to consider this aspect of trainable scene representations. We introduce an evaluation framework to quantify how well current state-of-the-art methods remove objects from scenes. Our evaluation combines four complementary metrics based on semantics (equation 1), segmentation (equation 2, equation 3) and depth (equation 4), capturing whether removed objects remain detectable at different levels of granularity. Experiments on indoor and outdoor scenes show that the proposed metrics are consistent in their ranking of the evaluated methods.

To complement our evaluation, we introduce Remove360, a new dataset, a dataset of diverse indoor and outdoor scenes with real pre-/post-removal captures, and corresponding object masks serving as ground truth. Unlike existing datasets such as 360-USID Wu et al. (2025), centered on staged single-object removals, Remove360 features multi-object, naturalistic scenes that better reflect real-world complexity and expose challenging residual artifacts. Initial experiments show that current state-of-the-art 3D removal methods often fail to generalize to Remove360, underscoring the open challenges and the relevance of this benchmark for future research.

In summary, our contributions are as follows: i) We propose an evaluation that measures how well scene removal operations remove objects in the context of privacy. To the best of our knowledge, this is the first work to explore this aspect; ii) We define quantitative metrics that support this evaluation and demonstrate their consistency and reliability across state-of-the-art methods for trainable 3D scenes; iii) We release a new dataset of real-world indoor and outdoor scenes with pre-/post-removal images, and masks of the removed objects. The dataset reveals failure cases in state-of-the-art methods—such as residual artifacts, incomplete removal, and over-smoothing—that are not exposed in existing benchmarks. These challenges make it a valuable resource for advancing research on robust, privacy-aware scene editing.

## 2 RELATED WORK

**3D reconstruction from images** builds 3D models that capture the scene's geometry and appearance. Popular scene representations are point clouds (Schonberger & Frahm, 2016; Yunhan Yang & Liu, 2023; Huang et al., 2024; Peng et al., 2023; Liu et al., 2024a; Yin et al., 2024), meshes (Schönberger et al., 2016; Furukawa & Ponce, 2009; Lazebnik et al., 2001; Kundu et al., 2020), and, more recently, Neural Radiance Fields (NeRFs) (Mildenhall et al., 2020; Barron et al., 2022; Reiser et al., 2021; Müller et al., 2022; Chen et al., 2024a; Kulhanek & Sattler, 2023; Martin-Brualla et al., 2021) and 3D Gaussian Splatting (3DGS) (Kerbl et al., 2023; Lin et al., 2024; Yu et al., 2024; Zhang et al., 2024; Kulhanek et al., 2024; Chen et al., 2024c; Wang et al., 2024).

NeRFs represent the scene implicitly with a colored volumetric field: for each 3D point in the space, an MLP outputs a volumetric density and a view-dependent color value. In 3DGS (Kerbl et al., 2023),

the scene is represented explicitly with a set of 3D Gaussians with learnable parameters (positions, orientation, scale, opacity, view-dependent color), generating a rendering similar to the original view.

**Linking 3D reconstructions and semantics.** NeRFs and 3DGS can easily be extended to embed semantic features that can be prompted via natural language Kerr et al. (2023); Tschernezki et al. (2022), pixel locations Kerr et al. (2023); Tschernezki et al. (2022) or semantic labels Ye et al. (2024a). The prompt allows humans to search for elements in the 3D reconstruction and then edit that location.

In NeRFs, the MLP is extended to output semantic features that are rendered in a differential manner and supervised with off-the-shelf 2D semantic features. Examples are NeRFs extended with text features Wang et al. (2022); Mirzaei et al. (2022) like CLIP (Radford et al., 2021), text and semantic features Kerr et al. (2023); Kobayashi et al. (2022), and unsupervised features Tschernezki et al. (2022) like DINO (Caron et al., 2021; Oquab et al., 2024). Prompting in reconstruction means locating the NeRF features most similar to the prompt. The feature field can also be trained to render segmentation masks consistent with 2D masks Zhi et al. (2021); Vora* et al. (2022); Cen et al. (2023b); Liu et al. (2024b) derived with semantic models Chen et al. (2017); Graham & Van der Maaten (2017); Li et al. (2022), panoptic models Siddiqui et al. (2023); Bhalgat et al. (2024), foundation models Ravi et al. (2024); Kirillov et al. (2023); Zou et al. (2023); Li et al. (2024); Liu et al. (2022), or user annotations Mirzaei et al. (2023). The prompt search then looks for the features associated with a given semantic label.

Although these methods perform well in locating prompted elements, the implicit nature of NeRFs makes it complex to associate their location with a set of abstract MLP parameters to edit. Hence, editing the reconstruction is not as straightforward as in 3DGS (Kerbl et al., 2023) representations that are explicit: removing a prompted element amounts to deleting the 3D Gaussians at its location. 3DGS can also be embedded with semantic features Cen et al. (2023a); Choi et al. (2024b); Jain et al. (2024); Wu et al. (2024b); Gu et al. (2024); Zhou et al. (2024); Wu et al. (2025), text features Shi et al. (2024); Liao et al. (2024); Qin et al. (2024), and unsupervised features Zuo et al. (2024). This motivates this paper to focus on 3D reconstructions represented with 3DGS.

**Evaluating 3D reconstruction approaches.** 3D reconstructions are evaluated based on the accuracy of the 3D geometry and whether the color renderings and the semantic (features) renderings are similar to those in the training views. Often, scene operations, such as editing and removal, are reported only as illustrative examples with qualitative results Ye et al. (2024a); Zhou et al. (2024); Gu et al. (2024); Wu et al. (2024b); Jain et al. (2024); Choi et al. (2024b). Some works Cen et al. (2023a); Choi et al. (2024b); Mirzaei et al. (2023) report a quantitative evaluation of 'background' removal by comparing the renderings of the searched element against its appearance in the original view. This paper addresses a different problem - are there residuals of the removed object present in the scene, if so, can they still be recognized or reasoned over? We propose an evaluation framework to answer this question in a quantitative and automatic manner. To the best of our knowledge, there is no previous work that addresses such a question.

**Privacy challenges.** The use of extensive public data in the recent scientific breakthroughs Rombach et al. (2022); Saharia et al. (2022); Brooks et al. (2024); Achiam et al. (2023) has drawn attention to the protection of user data in the research community Raina et al. (2023); Speciale et al. (2019); Pittaluga et al. (2019); Chelani et al. (2023); Moon et al. (2024); Nasr et al. (2023), in companies Rubinstein & Good (2013); Grynbaum & Mac (2023), and governments Illman & Temple (2019); Voigt & Von dem Bussche (2017).

This challenge will grow as the deployment in households of new types of sensors, eg. Augmented Reality / Virtual Reality (AR / VR) glasses (spe; Engel et al., 2023; xre), and autonomous systems will become the norm. An efficient way to make systems privacy-preserving is to consume data that has already been made privacy-preserving by the user. One relevant line of work proposes anonymizing the images Liu et al. (2024c); Weder et al. (2023) with inpainting before the scene reconstruction rather than editing the reconstruction later.

However, this method is more computationally complex: it involves editing many images as opposed to a single reconstruction and can introduce artifacts in the image that reduce the quality of the reconstruction. Hence, operating on the 3D model offers privacy at a reasonable computational cost and better reconstruction quality.

## 3    METRICS DEFINITION

We evaluate whether object removal in 3DGS (Kerbl et al., 2023) leaves identifiable traces of the removed content, with a focus on privacy when sharing scene representations. In this context, an element is private if it cannot be identified (Illman & Temple, 2019; Voigt & Von dem Bussche, 2017; Raina et al., 2023). Our metrics assess whether removed elements remain identifiable, assuming access to the ground-truth mask of the target object and focusing on changes within this region.

### 3.1    SEMANTIC OBJECT RECOGNITION

Semantic segmentation identifies objects by classifying each pixel into categories (Chen et al., 2017; Graham & Van der Maaten, 2017; Li et al., 2022). Here, it is used to test whether an object can still be recognized after removal by rendering the scene from multiple views and evaluating a segmentation model on those renderings (see Fig. 2). Comparing the segmentation performance before and after removal provides information on the removal quality. A drop in performance indicates that the object is removed. We thus define the semantic recognition metric as the segmentation's performance gap on the renderings before and after removal. The semantic segmentation is evaluated with the standard Intersection over Union (IoU) Chen et al. (2017); Badrinarayanan et al. (2017) that measures how well the estimated semantic mask overlaps with the ground-truth mask.

Specifically, $IoU_{pre}$ and $IoU_{post}$ are computed on the predicted semantic masks from renderings before and after removal. To reduce false positives, we only keep predictions overlapping with the ground-truth mask. The $IoU_{drop}$ is defined as:

$$IoU_{drop} = IoU_{pre} - IoU_{post}, \tag{1}$$

ranging from -1 to 1, with higher values ($\uparrow$) indicating better removal.

A low absolute value of the $IoU_{drop}$ implies $IoU_{post} = IoU_{pre}$, which can be interpreted in two ways. (1) Both $IoU_{post,pre}$ are high, so the object is recognized even after removal (failure). (2) Both $IoU_{post,pre}$ are low, meaning the model could not segment the object even in the original scene. No conclusion about removal quality can be drawn.

To handle this ambiguity, we add a complementary semantic metric defined in the next section. Still, $IoU_{drop}$ is useful on its own as a warning signal, especially in interactive systems where human oversight is possible, e.g., active labeling. In the experiments, we also report a more intuitive metric, the performance of the segmentation after removal, and analyse its correlation $IoU_{drop}$. We define the accuracy $acc_{seg}$ as the ratio of images after removal in which the semantic element is not recognized anymore. The element is not recognized if $IoU_{post}$ is smaller than a given threshold $\xi_{IoU}$, and we report this metric over multiple thresholds (see Supp. Tab. 6b, 9).

$acc_{seg}$ ranges from 0 to 1 and the higher $acc_{seg}$, the better the object removal, as indicated by the $\uparrow$.

$$acc_{seg,\xi_{IoU}} = \frac{\|\text{\# images with } IoU_{post} < \xi_{IoU}\|}{\|\text{\# images }\|} \tag{2}$$

### 3.2    ANYTHING RECOGNITION

The previous sections defined the $IoU_{drop}$ that can be interpreted in two ways when it is low. To address such ambiguity, we complement the $IoU_{drop}$ with a second semantic metric based on finer segmentations, i.e., object parts or instances instead of semantic categories. These segmentations are derived with the foundational SegmentAnything (SAM) Kirillov et al. (2023); Ravi et al. (2024), a prompt-based segmentation model that can be prompted with image locations, bounding boxes, or texts. The model can also operate without prompts, which results in a set of masks that cover all the semantic elements in the image (see Fig. 3).

In this evaluation, we assess whether an element has been removed by comparing the SAM Kirillov et al. (2023); Ravi et al. (2024) masks of the scene renderings before and after removal (in case we do not have ground truth after removal), and the scene renderings with ground truth after removal. When the object is removed or partially removed, SAM segments what is behind the object so the masks should change (see rows in Fig. 3).

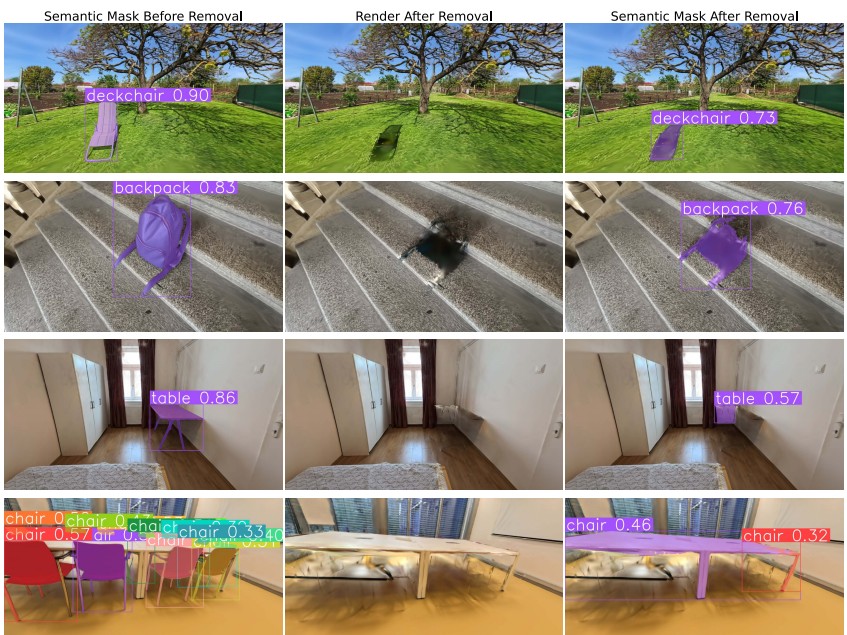

Figure 2: **Semantic segmentation changes before and after removal on Remove360.** Left-right: GroundedSAM2 Kirillov et al. (2023); Liu et al. (2023); Ren et al. (2024) overlay on the rendering before removal, rendering after removal, overlay after removal. These semantic masks are used to calculate change in semantic segmentation in equation 1 and its accuracy equation 2. Rows: Different object removals. Even though the object can not be recognized by a human, the segmentation model still finds it. One explanation can be that the pixel distribution on the edited area still exhibits patterns characteristic of the object, similar to what occurs in adversarial attacks.

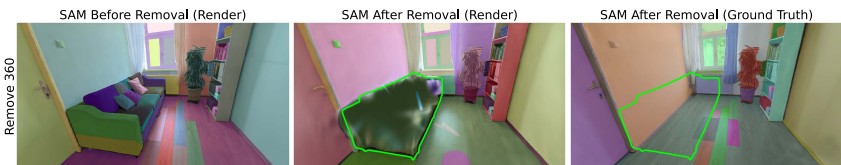

Figure 3: **SAM Kirillov et al. (2023); Ravi et al. (2024) mask comparison on Remove360.** Object removal alters SAM masks, and smaller changes relative to ground-truth masks indicate better removal. These differences are used to compute the similarity score equation 3. Left to right: SAM overlay before removal, after removal, and ground-truth with the object mask (green outline).

We next define $\text{sim}_{\text{SAM}}$ that measures the similarity between two sets of SAM Kirillov et al. (2023); Ravi et al. (2024) masks based on how well they spatially overlap. We first match the masks that overlap the most between the two sets. Then $\text{sim}_{\text{SAM}}$ is the average overlap between the mask matches. We enforce a 1-to-1 matching, i.e., a mask in one set is matched to at most one mask in the other. We do so by defining that two masks match if they overlay, and if one mask gets matched to more than one, we keep the match that leads to the maximum overlay over all matches. This is derived by solving an assignment problem that maximizes the overlay over all matches with the Hungarian algorithm Munkres (1957).

More formally, let $A = (a_i)_{i \in [1,N]}$ and $B = (b_j)_{j \in [1,M]}$ be the sets of comparing SAM masks, and $(a_k, b_k)_{k=1,K}$ be the $K$ matching masks. The similarity between these two sets is:

$$\text{sim}_{\text{SAM}} = \frac{\sum_{k=1}^{K} \text{IoU}(a_k, b_k)}{\max(N, M)} \quad (3)$$

$\text{sim}_{\text{SAM}}$ lies in $[0, 1]$. Based on set of comparing mask, we aim for higher $\text{sim}_{\text{SAM}}$ score, expecting masks to be more similar, or lower score, expecting masks to be less similar. Having ground truth after removal, comparison between after removal mask with the ground truth mask should yield high

Depth Before Removal          Depth After Removal          Depth Difference with Object Contours

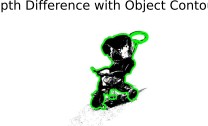

Figure 4: **Depth changes before and after removal.** Left to right: rendered depth before removal, rendered depth after removal, thresholded depth difference and ground-truth outline of the object to be removed in green. This depth difference is used for evaluation in equation 4. Results on Remove 360 showing localized changes in the depth maps, indicating not fully inaccurate object removal, somewhere over removed other under removed.

score, indicating no visual difference in the rendering, as indicated by the ↑. When comparing masks of renderings before and after removal, the score should be be lower, the less similar the masks are, hence the better the removal, as indicated by the ↓.

Note that we normalize the score with the highest number of masks $\max(N, M)$ instead of the number of mask matches $K$. We do so to account not only for the difference in overlay (in the numerator) but also for the difference in the number of masks. To reflect the changes related to the removed object, the metric is derived only over the masks that overlay with the object with an IoU of at least 0.1.

### 3.3 SPATIAL RECOGNITION

We complete the previous metrics with one that depends only on the 3D scene before and after removal, hence increasing the robustness of the evaluation against possible errors in the segmentations.

Inspired by recent works in scene change detection (Adam et al., 2022), we measure how well an object is removed based on the changes in the rendered depths before and after removal: a strong change in the depth maps indicates a change in the scene.

Hence, if the depth of the object changes enough, then the object is well removed.

More formally, we report the ratio of the object's pixels which depth changes by more than a threshold $\xi_{\text{depth}}$. The threshold $\xi_{\text{depth}}$ is derived automatically with Generalized Histogram Thresholding (Barron, 2020) on the histogram of depth differences over the whole image (see Fig. 4).

The depth maps are derived from the scene's rendering before and after removal so they have consistent scales. The defined ratio can be interpreted as the accuracy in depth change and is noted $\text{acc}_{\Delta\text{depth}}$:

$$\text{acc}_{\Delta\text{depth}} = \frac{\#\text{object pixel with depth change} > \xi_{\text{depth}}}{\#\text{object pixels}} \quad (4)$$

The object's pixel locations are specified by the object's mask in the image that we assume is available.

## 4 OBJECT REMOVAL DATASET REMOVE360

To facilitate the evaluation of object removal methods, we introduce Remove360, a new dataset featuring RGB images of scenes both before and after object removal, along with accurate object masks. Unlike 360-USID Wu et al. (2025), which focuses on single-object, carefully aligned captures with one reference view, Remove360 targets complex, real-world scenarios with multiple interacting objects and rich scene context. This makes it a more challenging and realistic benchmark for object removal.

**Dataset Composition.** Remove360 contains 11 scenes: 5 indoor and 6 outdoor (see Fig. 5). Each scene includes: (1.) Training views: RGB images and object masks before removal. (2.) Testing views: RGB images of the same scene after removal, providing ground truth for novel view synthesis and residual detection. Each scene contains between 150 and 300 training views, with a comparable number of testing views. The number of removed objects per scene ranges a single item (e.g., backpack, bicycle), to pairs (e.g., white plastic chairs), and up to several objects (e.g., pillows

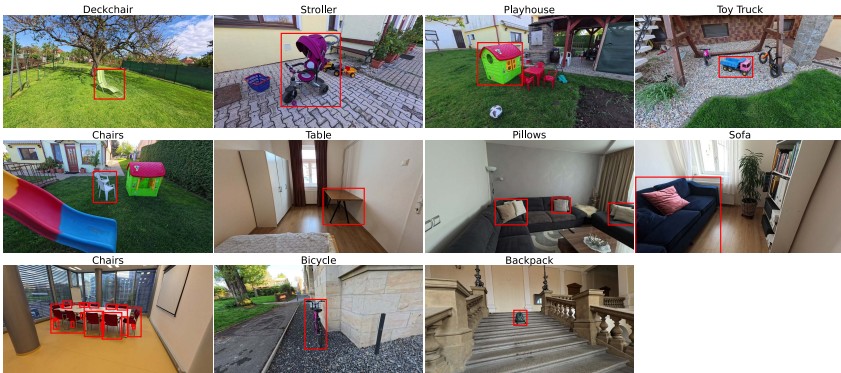

Figure 5: **Overview of the Remove360 dataset.** Samples from 11 scenes (5 indoor, 6 outdoor) with varied object counts, layouts, and interactions. Removed objects are shown with bounding boxes.

or multiple chairs in a conference room). The removed objects cover a wide range of physical characteristics, varying in size—from small (e.g., toy truck, backpack) to large (e.g., sofa, table)—and shape, from compact (e.g., backpack, playhouse) to complex shapes (e.g., deckchair, bicycle). Object masks were initially generated using SAM Kirillov et al. (2023) and subsequently refined by manual annotation. To measure the impact of inaccuracies in the ground truth masks on the results of the evaluation process, we performed a mask erosion and dilation analysis. The analysis validated that the evaluation truly focuses on removal fidelity and is robust to small boundary inaccuracies (see Supp. A.3).

**Dataset Collection Protocol.** We recorded 4K 60fps videos using the Insta360 AcePro camera. Each scene was captured over 4–6 minutes. We selected the sharpest frame per second using the variance of the Laplacian method. Camera poses were recovered using the hloc structure-from-motion pipeline, with SuperPoint and LightGlue for feature matching (see Supp. A.2).

## 5 EXPERIMENTS

**Methods.** We evaluate five publicly available methods for object removal. To ensure fair comparison, no additional inpainting or refinement is applied after removal. We focus on Gaussian Splatting due to its explicit and interpretable 3D representation, which allows direct manipulation and evaluation of individual scene components. However, our evaluation framework is not limited to Gaussian Splatting—any 3D representation can be evaluated as long as renderings and depth maps before and after removal are available.

*Feature3DGS (FGS)* Zhou et al. (2024) distills the LSEG (Li et al., 2022) semantic features aligned with CLIP's text features (Radford et al., 2021). FGS is prompted with a tuple of text entries: one positive query is associated with the object of interest and the others are negative queries. The search compares the Gaussians' feature with the features of each text entry, and their similarity is normalized.

*GaussianGrouping (GG)* Ye et al. (2024a) distills SAM (Kirillov et al., 2023) features that operate at a finer granularity than LSEG (Li et al., 2024). Also, GG enforces spatial consistency between semantically similar Gaussians so that close-by Gaussians have similar features. A Gaussian is removed if its feature is associated with the prompted instance label. Post-processing then removes all Gaussians within the convex hull of the removed Gaussians.

*SAGS* Hu et al. (2024) is a training- and feature-free method that removes Gaussians based on their projection overlap with 2D object masks across views. It estimates a removal probability for each Gaussian. The 3D center of the Gaussian is projected on the images and the removal probability is the ratio of images in which the projections land on the object's location. When assigning Gaussians to the object does not account for the Gaussian's opacity, which may lead to over-removal.

*GaussianCut (GC)* Jain et al. (2024) leverages the spatial and color correlations between Gaussians. It models a trained 3DGS Kerbl et al. (2023) scene as a graph of Gaussians and removes them through graph-cut optimization using 2D object mask prompts, without features or training. The

| Scene- | IoU_drop ↑ | | | | | acc_seg, IoU_post < 0.5 ↑ | | | | | acc_Δdepth ↑ | | | | | sim_SAM ↑ | | | | |
|---|---|---|---|---|---|---|---|---|---|---|---|---|---|---|---|---|---|---|---|---|
| Object | FGS | GG | SAGS | GC | AF | FGS | GG | SAGS | GC | AF | FGS | GG | SAGS | GC | AF | FGS | GG | SAGS | GC | AF |
| Backyard- Deckchair | * | * | * | **0.85** | 0.84 | * | * | * | **0.99** | **0.99** | * | * | * | **0.67** | 0.65 | * | * | * | **0.56** | 0.54 |
| Backyard- Chairs | * | * | * | 0.85 | **0.87** | * | * | * | **1.00** | **1.00** | * | * | * | **0.76** | 0.67 | * | * | * | **0.83** | 0.62 |
| Backyard- Stroller | * | * | * | **0.92** | 0.91 | * | * | * | **1.00** | **1.00** | * | * | * | **0.89** | 0.73 | * | * | * | **0.85** | 0.72 |
| Backyard- Playhouse | * | * | * | 0.95 | **0.97** | * | * | * | **1.00** | **1.00** | * | * | * | **0.92** | 0.87 | * | * | * | **0.50** | 0.49 |
| Backyard- Toy Truck | * | * | * | **0.95** | 0.93 | * | * | * | 0.99 | **1.00** | * | * | * | **0.73** | 0.64 | * | * | * | **0.22** | 0.20 |
| Bedroom- Table | * | * | * | **0.91** | **0.91** | * | * | * | 0.98 | **1.00** | * | * | * | 0.57 | **0.58** | * | * | * | **0.48** | 0.44 |
| Living Room- Pillows | * | * | * | 0.62 | **0.76** | * | * | * | 0.77 | **0.88** | * | * | * | **0.53** | 0.51 | * | * | * | **0.19** | 0.18 |
| Living Room- Sofa | * | * | * | 0.57 | **0.62** | * | * | * | 0.50 | **0.64** | * | * | * | 0.62 | 0.62 | * | * | * | **0.17** | 0.13 |
| Office- Chairs | * | * | * | **0.69** | 0.64 | * | * | * | **0.85** | 0.76 | * | * | * | **0.91** | 0.82 | * | * | * | **0.34** | 0.33 |
| Park- Bicycle | * | * | * | **0.95** | **0.95** | * | * | * | 0.99 | **1.00** | * | * | * | **0.91** | 0.80 | * | * | * | **0.68** | 0.48 |
| Stairwell- Backpack | * | * | * | **0.89** | 0.82 | * | * | * | **0.93** | 0.85 | * | * | * | **0.73** | 0.65 | * | * | * | **0.37** | **0.37** |

(a) Remove360 dataset evaluation results.

| Scene- | IoU_drop ↑ | | | | | acc_seg, IoU_post < 0.5 ↑ | | | | | acc_Δdepth ↑ | | | | | sim_SAM ↓ | | | | |
|---|---|---|---|---|---|---|---|---|---|---|---|---|---|---|---|---|---|---|---|---|
| Object | FGS | GG | SAGS | GC | AF | FGS | GG | SAGS | GC | AF | FGS | GG | SAGS | GC | AF | FGS | GG | SAGS | GC | AF |
| Counter- Baking Tray | 0.34 | 0.53 | 0.10 | **0.62** | 0.60 | 0.78 | 0.91 | 0.48 | **0.99** | 0.96 | **0.99** | 0.96 | 0.21 | 0.98 | 0.76 | **0.21** | 0.35 | 0.71 | 0.35 | 0.37 |
| Plant | 0.75 | 0.84 | 0.03 | 0.86 | **0.87** | **1.00** | **1.00** | 0.17 | **1.00** | **1.00** | **1.00** | **1.00** | 0.01 | 0.99 | 0.74 | 0.13 | **0.12** | 0.85 | 0.13 | 0.13 |
| Gloves | 0.01 | 0.60 | 0.10 | 0.60 | 0.65 | 0.28 | 0.84 | 0.34 | 0.83 | 0.89 | 0.01 | **1.00** | 0.55 | **1.00** | 0.74 | 0.99 | **0.12** | 0.56 | 0.16 | 0.17 |
| Egg Box | 0.08 | **0.63** | 0.56 | 0.62 | 0.63 | 0.20 | **1.00** | 0.96 | 0.99 | 0.99 | 0.06 | **1.00** | 0.86 | **1.00** | 0.39 | 0.84 | **0.15** | 0.47 | 0.19 | 0.79 |
| Room- Plant | **0.53** | 0.26 | 0.17 | **0.53** | 0.23 | **1.00** | 0.80 | 0.72 | **1.00** | 0.96 | **0.97** | 0.70 | 0.33 | 0.99 | 0.43 | 0.22 | 0.33 | 0.57 | 0.14 | **0.07** |
| Slippers | 0.00 | **0.82** | 0.25 | 0.48 | 0.42 | 0.06 | 0.02 | **0.83** | 0.28 | 0.44 | 0.00 | **1.00** | 0.91 | 0.98 | 0.38 | 1.00 | **0.05** | 0.35 | 0.35 | 0.15 |
| Coffee table | 0.57 | **0.86** | 0.00 | **0.86** | 0.55 | 0.62 | **0.99** | 0.09 | **0.99** | 0.98 | 0.67 | 0.89 | 0.06 | **0.99** | 0.53 | 0.26 | 0.08 | 0.86 | 0.07 | **0.05** |
| Kitchen- Truck | 0.62 | 0.61 | 0.67 | 0.66 | **0.95** | 0.95 | 0.92 | 1.00 | 0.99 | **1.00** | 0.96 | **1.00** | **1.00** | 0.92 | 0.86 | 0.35 | 0.17 | 0.22 | **0.08** | 0.19 |
| Garden- Table | 0.67 | 0.48 | 0.81 | 0.86 | **0.90** | 0.70 | 0.54 | 0.88 | 0.95 | **1.00** | 0.99 | **1.00** | 0.98 | **1.00** | 0.57 | 0.11 | 0.14 | **0.04** | 0.06 | 0.10 |
| Ball | 0.00 | 0.16 | 0.41 | **0.42** | **0.42** | 0.94 | **1.00** | **1.00** | **1.00** | **1.00** | 0.00 | 0.60 | 0.60 | 0.53 | 0.47 | 0.59 | **0.01** | 0.21 | 0.37 | 0.13 |
| Vase | 0.79 | 0.64 | 0.96 | **0.97** | 0.97 | 0.89 | 0.79 | **1.00** | **1.00** | **1.00** | 0.99 | **1.00** | 0.96 | **1.00** | 0.92 | 0.12 | **0.10** | 0.11 | 0.11 | 0.11 |

(b) Mip-NERF360 Barron et al. (2022) dataset evaluation results.

Table 1: **Object removal evaluation with the proposed metrics on two datasets.** The four metrics measure changes in semantics and depth before and after removal: $IoU_{drop}$ measures the drop in semantic segmentation after removal, $acc_{seg, \xi_{IoU}}$ measures the ratio of images after removal in which the semantic element is not recognized anymore, having $IoU_{post} < 0.5$, $acc_{\Delta depth}$ captures changes in the depth maps, and $sim_{SAM}$ quantifies difference in the SAM Kirillov et al. (2023) masks. The **best** and second-best are highlighted each metrics. (a) On the Remove360, GaussianCut (GC) Jain et al. (2024) outperforms AuraFusion (AF) Wu et al. (2025), especially in the instance segmentation similarity $sim_{SAM}$. (b) On the Mip-NERF360 dataset Barron et al. (2022) GC and Gaussian Groupping (GG) Ye et al. (2024a) mostly outperform Feature3DGS (FGS) Zhou et al. (2024) and SAGS Hu et al. (2024), while AF achieves the best results in semantic segmentation drop ($IoU_{drop}$). In the Mip-NERF360 we do not have ground truth after removal, therefore we compare instance segmentation of the renders before and after removal, expecting to see lower similarity $sim_{SAM}$ score.
removal probability value is initialized by lifting the 2D prompt mask to 3D and refined via graph-cut optimization where the unary term represents the likelihood of the Gaussian to be removed and the binary term measures the color similarity and spatial distance between two Gaussians.

*AuraFusion360 (AF)* Wu et al. (2025) is a training-based method using multi-view RGB images and object masks. It removes objects using depth-aware mask generation to handle occlusions.

**Datasets.** Methods are evaluated on our Remove360 dataset and the Mip-NeRF360 dataset Barron et al. (2022). Since Mip-NeRF360 Barron et al. (2022) lacks semantic masks, we generate pseudo-ground-truth masks using SAM Kirillov et al. (2023) and human annotations (see Supp. A.3). We use the Mip-NeRF360 dataset as it has been commonly used by the approaches we consider in this work.

**Implementation details.** Due to memory constraints, we use 100–150 images per used scene from Mip-NeRF360 (kitchen, counter, room, garden), which is sufficient per prior studies (Jain et al., 2024; Zhou et al., 2024). All methods are trained on the same image subsets and 3D point clouds reconstructed using Remove360 scenes are used in full. For evaluation, we apply GroundedSAM2 Kirillov et al. (2023); Liu et al. (2023); Ren et al. (2024) to compute metrics based on semantic segmentation, and SAM Kirillov et al. (2023) for instance segmentation metrics.

## 5.1 RESULTS

**Methods comparison.** Tab. 1 reports the considered approaches under our proposed metrics. While methods perform well overall, closer inspection and visualizations show persistent semantic residuals, indicating that current removal methods remain imperfect (see Supp. A.1).

On Remove360, GaussianCut (GC) Jain et al. (2024) outperforms AuraFusion (AF) Wu et al. (2025), particularly in instance segmentation similarity (simSAM), suggesting more accurate and complete object removal. AF benefits from training on multi-view masks, however, this characteristic did not translate to ideal performance, where removing objects using AF resulted in less similar instance segmentations after the removal, lower $\text{sim}_{SAM}$. $\text{IoU}_{drop}$, $\text{sim}_{SAM}$, and $\text{acc}_{seg}$ appear to correlate, confirming that greater semantic change and segmentation similarity with ground truth novel view after removal, results in better removal. However, some scenes (e.g., living room, office) still show residual traces, and depth accuracy ($\text{acc}_{\Delta\text{depth}}$) remains low—indicating limited depth modification. Notably, several methods (Ye et al., 2024a; Zhou et al., 2024; Hu et al., 2024) fail on Remove360, unable to detect removed objects, showcasing its difficulty. This suggests that these methods might not be altering the depth information of the removed objects as effectively as they are altering the semantic and instance segmentation. On the Remove360, methods such as (Jain et al., 2024; Wu et al., 2025) are trained and evaluated on the full Remove360 dataset. In contrast, other methods (Ye et al., 2024a; Zhou et al., 2024; Hu et al., 2024) could not be evaluated on Remove360, as they failed to learn meaningful object representations and were thus unable to detect removed objects. As a result, their evaluation metrics remain identical before and after object removal, indicating failure to detect changes in Tab. 1.

For the Mip-Nerf360 Barron et al. (2022), the methods' ranks remain stable across the three metrics: AF Wu et al. (2025), GC Jain et al. (2024), and GG Ye et al. (2024a) lead across metrics. In contrast, FGS Zhou et al. (2024) under performs, likely due to prompt sensitivity, and SAGS Hu et al. (2024) shows high variance, performing better in spatially distinct objects-centric scenes (garden, kitchen). Mip-NeRF360 Barron et al. (2022) lacks post-removal ground truth, so only $\text{sim}_{SAM}$ between pre- and post-removal renderings is available, therefore we expect to see lower similarity.

Importantly, Remove360 introduces real-world challenges with paired pre/post-removal images and masks, enabling direct measurement of semantic residuals and post-removal segmentation. Unlike MipNeRF360, it supports ground-truth-based evaluation and reveals generalization gaps, offering a more rigorous benchmark for future methods.

**Qualitative Results.** Figs. 2, 3, and 4 show renderings of the evaluated methods before and after removal. Fig. 2 illustrates an interesting case where the object is not visible to humans anymore, yet GroundedSAM2 Kirillov et al. (2023); Liu et al. (2023); Ren et al. (2024) finds the object. Our dataset was not available until now, and thus is not part of GroundedSAM2's training data. This suggests that barely visible information about the object can remain in the scene, even when the removal is successful to the human eye, and that the proposed metrics can detect such scenarios. This opens interesting future directions on whether a network could be trained to invert the object removal from invisible pixel information and how to prevent it.

Fig. 3 shows the distribution of SAM Kirillov et al. (2023) masks on the rendering before and after removal, compared to ground truth instance segmentation after removal. It provides a visual intuition on how $\text{sim}_{SAM}$ behaves: successful 'sofa' removal should reveal new segments behind it. Fig. 4 provides a visual intuition for $\text{acc}_{\Delta\text{depth}}$. Successful removal causes depth differences localized in the object area (object mask outlined in green). More visualizations and quantitative results are provided in Supplementary (for Remove360 see B, and for Mip-NERF360 see C).

**Limitations.** The metrics rely on off-the-shelf semantic segmentation models that can introduce errors. Although introducing redundancy between the metrics alleviates this issue, it does not fully address it, which calls for further research on robust metrics for the evaluation of object removal.

## 6 CONCLUSION

We introduce a novel evaluation framework for assessing object removal in 3D Gaussian Splatting, targeting privacy-preserving scene representations. Our metrics combine off-the-shelf semantic models and depth reasoning to quantify whether removed objects leave detectable residuals. Experiments on state-of-the-art methods reveal persistent semantic traces, underscoring key limitations in current approaches. To enable rigorous, ground-truth-based evaluation, we release Remove360, a challenging real-world dataset with paired pre- and post-removal images and object masks.

We hope this work lays the foundation for future research in privacy-preserving 3D scene manipulation, where removal operations leave no recoverable trace.

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
