# SUPPLEMENTARY MATERIAL

The supplementary material provides additional implementation details, evaluation metrics, and extended experimental results supporting the main paper. The evaluation framework is available at https://github.com/spatial-intelligence-ai/Remove360.git. Data are available at https://huggingface.co/datasets/simkoc/Remove360.

Section A describes the implementation details. Subsection A.1 presents the parameter settings for the compared methods: Feature3DGS Zhou et al. (2024), SAGS Hu et al. (2024), GaussianGrouping Ye et al. (2024a), Gaussian Cut Jain et al. (2024), and Aura Fusion Wu et al. (2025). Subsection A.2 describes dataset processing protocol used for Remove360. Subsection A.3 describes how the pseudo-groung-truth masks for Mip-NERF360 were obtained. Subsection A.4 describes the evaluation metrics.

Section B presents additional results for the Remove360 Dataset. Subsection B.1 presents quantitative results. Subsection B.2 presents qualitative results.

Section C presents additional results for the Mip-NERF360 Dataset. Subsection C.1 presents quantitative results. Subsection C.2 presents qualitative results.

## A  IMPLEMENTATION DETAILS

### A.1  COMPARED METHODS

Official implementations provided by the respective authors are used with the following settings.

**Feature3DGS (FGS)** Zhou et al. (2024) is prompted with a tuple of text entries: one positive text prompt is associated with the object of interest and the others are negative text prompts. The search compares the Gaussians' feature with the features of each text entry, and their similarity is normalized with softmax. A Gaussian is removed if the similarity between the Gaussian's feature and the prompt feature is higher than a threshold. We set this similarity threshold to $0.4$ for all scenes and objects.

The following negative prompts are used in Mip-NERF360, per scene: Garden: {grass, sidewalk, tree, house}, Room: {sofa, rug, television, floor}, Kitchen: {rug, table, chair}, Counter: {oranges, wooden rolling pin, coconut oil}.

The following negative prompts are used in Remove360, per scene: Backyard- Deckchair : {tree, grass, sky} Backyard- Chairs : {house, sidewalk, grass, plant, sky} Backyard- Stroller : {sidewalk, grass, plant, bench} Backyard- Playhouse : {tree, grass, plant, sky} Backyard- Toy Truck : {fence, grass, plant, dirt} Bedroom- Table : {cabinet, floor, bed, wall} Living Room- Pillows : {sofa, armchair, wall, rug, curtain} Living Room- Sofa : {plant, wall, floor, curtain, shelf} Office- Chairs : {table, floor, window, wall} Park- Bicycle : {sidewalk, road, plant, sky, wall} Stairwell- Backpack : {stairs, wall, ceiling}

**SAGS** Hu et al. (2024) is a training-free and feature-free method, taking object masks as prompts. It estimates a removal likelihood for each Gaussian based on projective geometry. The 3D center of the Gaussian is projected on the images and the removal probability is the fraction of images in which the projections land in the object mask. The object masks therefore need to be available. A Gaussian is removed if its removal likelihood is higher than $0.7$.

**Gaussian Grouping** Ye et al. (2024a) takes SAM Kirillov et al. (2023) features and use them to assign a label for each Gaussian. After training, Gaussian is removed if its label is equal to selected label in config file for each scene. For the label training, SAM IoU prediction threshold is set to $0.8$. For Gaussian training, the default settings are used, densify_until_iter = 10000, num_classes = 256, reg3d_interval = 5, reg3d_k = 5, reg3d_lambda_val = 2, reg3d_max_points = 200000, reg3d_sample_size = 1000. For the object removal setting, the default number of classes of 256 and the removal threshold of $0.3$ are used.

**Gaussian Cut** Jain et al. (2024) is a feature-free and training-free method that leverages the spatial and color correlations between Gaussians. Given a trained 3DGS Kerbl et al. (2023) scene, it models the scene as a graph and determines which Gaussians should be removed via graph optimization. As for SAGS, the prompt is a set of object masks. The Gaussians define the nodes of the graph and are extended with a single parameter representing the probability of the Gaussian to be removed. The parameter is initialized by lifting the 2D prompt mask to 3D and refined via graph-cut optimization where the unary term represents the likelihood of the Gaussian to be removed and the binary term measures the color similarity and spatial distance between two Gaussians. The graph is built with the following parameters: each Gaussian is connected to its 10 nearest neighbors in 3D space- 'number of edges' per node is 10). The 'terminal clusters' define how foreground (source) and background (sink) labels are seeded in the graph-cut. Specifically, setting the 'terminal cluster source' = 5 and 'terminal cluster sink' = 5 mean that 5 clusters of Gaussians (likely foreground and background respectively) are selected to initialize the optimization. The 'leaf size' = 40 controls the granularity of the spatial clustering used to construct the graph efficiently. A 'foreground threshold' of 0.9 is applied when Gaussians connected the the object mask are visible in at least 90% of the masked views are considered for removal. The prompt is a set of multi-view masks associated with the object to be removed.

**Aura Fusion** Wu et al. (2025) jointly fuses 2D semantic masks with the 3D Gaussian representation. During training, it is supervised by either ground-truth masks or pseudo-ground-truth masks depending on the dataset. Training runs for 20,000 iterations and the object masks used as supervision are dilated with a kernel size of 10. The model learns to predict an object removal confidence for each Gaussian. To better handle occlusions and capture shape priors, a diffusion depth module is used to propagate 2D mask information into the 3D scene along view-dependent depth directions. At inference time, a Gaussian is removed if its predicted removal confidence exceeds a threshold of 0.6. An additional unseen object threshold of 0.0 is used to control background filtering, ensuring that only Gaussians with non-zero predicted relevance to the object are considered for removal.

## A.2 REMOVE360 DATASET PROCESSING PROTOCOL

**Camera Pose Estimation.** Camera poses and sparse scene geometry are reconstructed using the Hierarchical Localization Sarlin et al. (2019; 2020) (hLoc) pipeline. The steps are as follows:

1. Global Feature Extraction: Global descriptors are extracted using NetVLAD Arandjelović et al. (2016).

2. Local Feature Extraction and Matching: Local features are extracted using SuperPoint DeTone et al. (2018) (Aachen configuration) and matched using LightGlue Lindenberger et al. (2023) under the superpoint+lightglue configuration.

3. Image Pair Selection: Sequential image pairs are generated with a temporal overlap of 10 frames, with enabled quadratic overlap to match frames at exponentially increasing intervals. Loop closure detection is performed every 5th frame by retrieving the top 20 most similar images based on NetVLAD Arandjelović et al. (2016) descriptors.

4. Structure-from-Motion (SfM) Reconstruction: Sparse reconstruction is performed using COLMAP Schönberger & Frahm (2016); Schönberger et al. (2016) via the pycolmap interface. The RADIAL camera model is used. Camera parameters are used to undistort the input images, and both distorted and undistorted reconstructions are retained.

## A.3 OBJECT MASKS

**Pseudo-Ground-Truth Object Masks for Mip-NERF360.** Mip-NeRF360 Barron et al. (2022) does not provide ground-truth semantic masks necessary for our evaluation. To address this, pseudo-ground-truth masks are generated by applying SAM Kirillov et al. (2023) to each image, segmenting all objects, and selecting the masks corresponding to the target objects. When an object is covered by multiple overlapping segments, all relevant segments are combined to fully capture the object. These masks are sufficiently accurate for evaluation purposes. To ensure reliability, cases with incomplete segmentation from SAM are excluded from evaluation. However, such cases are rare.

| Scene | Object | Method | E 15 | E 10 | E 5 | Original | D 5 | D 10 | D 15 |
|-------|--------|--------|------|------|-----|----------|-----|------|------|
| Backyard | Deckchair | GC | **0.89** | **0.89** | **0.88** | **0.85** | **0.81** | **0.73** | **0.66** |
| | | AF | 0.88 | 0.88 | 0.87 | 0.84 | 0.79 | 0.71 | 0.65 |
| | Chairs | GC | 0.88 | 0.88 | 0.88 | 0.85 | 0.80 | 0.72 | 0.66 |
| | | AF | **0.89** | **0.89** | **0.89** | **0.87** | **0.82** | **0.75** | **0.69** |
| | Stroller | GC | **0.94** | **0.94** | **0.94** | **0.92** | **0.88** | **0.84** | **0.80** |
| | | AF | 0.93 | 0.93 | 0.93 | 0.91 | 0.87 | 0.83 | 0.79 |
| | Playhouse | GC | 0.95 | 0.95 | 0.95 | 0.95 | 0.93 | 0.92 | 0.90 |
| | | AF | **0.98** | **0.98** | **0.98** | **0.97** | **0.96** | **0.94** | **0.92** |
| | Toy Truck | GC | **0.96** | **0.96** | **0.96** | **0.95** | **0.91** | **0.87** | **0.83** |
| | | AF | 0.94 | 0.94 | 0.94 | 0.93 | 0.89 | 0.85 | 0.81 |
| Bedroom | Table | GC | **0.94** | **0.93** | **0.93** | **0.91** | **0.85** | **0.82** | **0.76** |
| | | AF | **0.94** | **0.93** | **0.93** | **0.91** | **0.85** | **0.82** | **0.76** |
| Living Room | Pillows | GC | 0.63 | 0.63 | 0.63 | 0.62 | 0.60 | 0.57 | 0.55 |
| | | AF | **0.78** | **0.78** | **0.77** | **0.76** | **0.73** | **0.70** | **0.67** |
| | Sofa | GC | 0.58 | 0.58 | 0.58 | 0.57 | 0.55 | 0.53 | 0.51 |
| | | AF | **0.63** | **0.63** | **0.63** | **0.62** | **0.59** | **0.57** | **0.54** |
| Office | Chairs | GC | **0.75** | **0.74** | **0.72** | **0.69** | **0.64** | **0.58** | **0.54** |
| | | AF | 0.71 | 0.69 | 0.67 | 0.64 | 0.59 | 0.54 | 0.50 |
| Park | Bicycle | GC | **0.97** | **0.97** | **0.96** | **0.95** | **0.91** | **0.86** | **0.82** |
| | | AF | **0.97** | **0.97** | **0.96** | **0.95** | **0.91** | **0.86** | **0.82** |
| Stairwell | Backpack | GC | **0.90** | **0.90** | **0.90** | **0.89** | **0.83** | **0.77** | **0.72** |
| | | AF | 0.84 | 0.84 | 0.84 | 0.82 | 0.77 | 0.70 | 0.65 |

Table 2: **IoU$_{drop}$ results for mask erosion (E) and dilation (D) analysis, Remove360 dataset.** Results show, how evaluation metric IoU$_{drop}$ changes when the ground truth mask is eroded or dilated by 5-15 pixels compared to the original mask used for Remove360 dataset. The **best** value between methods is highlighted for each object metric and mask state. Erosion consistently improves scores (Deckchair: $0.85 \rightarrow 0.89$), suggesting residuals remain near boundaries for both methods (which are then not taken into account when eroding the ground truth masks). Dilation lowers scores (Deckchair: $0.85 \rightarrow 0.66$), indicating inclusion of nearby artifacts like shadows. Thin objects (chairs, bike) are more sensitive, while larger ones (sofa, playhouse) are less affected. This confirms that our evaluation is not only robust, but object-aware, capturing residual traces at a fine-grained level.

**Ground-Truth Object Masks for Remove360.** Were initialized semantic masks using SAM, and then manually verified all masks, merging oversegmented regions. When needed, manually verified and refined by adding/removing pixels. Fewer than 10 images per scene (of up to 300) required edits. We estimate that the segmentations are accurate up to a pixel or two at the boundaries. No parts are missing and no unrelated parts are included in the masks that are used as ground truth. To measure the impact of inaccuracies in the ground truth masks on the results of the evaluation process, we performed a mask erosion and dilation analysis, see Tab. 2, 3, 4 We used OpenCV's Bradski (2000) morphological operation with an elliptical kernel. This expands or shrinks the masks by ±5, ±10, or ±15 pixels. The elliptical kernel avoids unrealistic boxy boundaries and approximates object contours more faithfully than a rectangular structuring element. Note that in our experience, the masks are more accurate than 5 pixels, i.e. the inaccuracies we observed are below 5 pixels on the boundary.

Dilation simulates potential over-segmentation or imprecise labeling, and allows us to measure how our metrics behave when neighboring context is included. As our results show, the metrics (IoU$_{drop}$ Tab. 2, acc$_{\Delta depth}$ Tab. 3, sim$_{SAM}$ Tab. 4) gradually degrade under dilation, indicating that they are sensitive to spatial precision but not overly affected by nearby irrelevant regions.

Conversely, erosion (shrinking the mask inward with the same elliptical kernel) helps isolate the object core, where removal is most likely to be clean. The consistent improvement in scores under erosion across multiple scenes validates that our evaluation truly focuses on removal fidelity and is robust to small boundary inaccuracies.

## A.4 METRICS

**Semantic Recognition.** If GroundedSAM2 Kirillov et al. (2023); Liu et al. (2023); Ren et al. (2024) fails to detect an object for a given prompt, no semantic mask is produced, and the semantic IoU is 0.

**Complementarity Analysis of the Metrics.** The ranking of the methods between the different metrics is mostly consistent, which is expected since they are all designed to measure the removal

| Scene | Object | Method | E 15 | E 10 | E 5 | Original | D 5 | D 10 | D 15 |
|-------|--------|--------|------|------|-----|----------|-----|------|------|
| Backyard | Deckchair | GC | **0.69** | **0.69** | **0.67** | **0.67** | **0.65** | **0.65** | **0.64** |
| | | AF | 0.67 | 0.66 | 0.65 | 0.65 | 0.65 | 0.59 | 0.55 |
| | Chairs | GC | **0.77** | **0.77** | **0.76** | **0.76** | **0.76** | **0.75** | **0.75** |
| | | AF | 0.67 | 0.67 | 0.67 | 0.67 | 0.65 | 0.62 | 0.58 |
| | Stroller | GC | **0.89** | **0.89** | **0.89** | **0.89** | **0.88** | **0.87** | **0.86** |
| | | AF | 0.72 | 0.73 | 0.73 | 0.73 | 0.72 | 0.70 | 0.67 |
| | Playhouse | GC | **0.92** | **0.92** | **0.92** | **0.92** | **0.92** | **0.91** | **0.91** |
| | | AF | 0.87 | 0.87 | 0.87 | 0.87 | 0.86 | 0.85 | 0.84 |
| | Toy Truck | GC | **0.73** | **0.73** | **0.73** | **0.73** | **0.73** | **0.72** | **0.71** |
| | | AF | 0.65 | 0.64 | 0.64 | 0.64 | 0.63 | 0.61 | 0.59 |
| Bedroom | Table | GC | **0.58** | **0.58** | 0.57 | 0.57 | 0.56 | **0.56** | **0.55** |
| | | AF | **0.58** | **0.58** | **0.58** | **0.58** | **0.57** | **0.56** | **0.55** |
| Living Room | Pillows | GC | **0.53** | **0.53** | **0.53** | **0.53** | **0.52** | **0.52** | **0.52** |
| | | AF | 0.51 | 0.51 | 0.51 | 0.51 | 0.51 | 0.51 | 0.50 |
| | Sofa | GC | **0.63** | **0.62** | **0.62** | **0.62** | **0.62** | **0.62** | **0.62** |
| | | AF | 0.62 | **0.62** | **0.62** | **0.62** | **0.62** | **0.62** | 0.61 |
| Office | Chairs | GC | **0.94** | **0.94** | **0.93** | **0.91** | **0.90** | **0.88** | **0.87** |
| | | AF | 0.83 | 0.83 | 0.82 | 0.82 | 0.81 | 0.79 | 0.78 |
| Park | Bicycle | GC | **0.91** | **0.91** | **0.91** | **0.91** | **0.90** | **0.90** | **0.89** |
| | | AF | 0.80 | 0.80 | 0.80 | 0.80 | 0.80 | 0.78 | 0.76 |
| Stairwell | Backpack | GC | **0.73** | **0.73** | **0.73** | **0.73** | **0.72** | **0.71** | **0.71** |
| | | AF | 0.66 | 0.66 | 0.66 | 0.65 | 0.65 | 0.62 | 0.60 |

Table 3: **acc$_{\Delta\text{depth}}$ results for mask erosion (E) and dilation (D) analysis, Remove360 dataset.** Results show, how evaluation metric acc$\Delta$depth changes when the ground truth mask is eroded or dilated by 5-15 pixels compared to the original mask used for Remove360 dataset. The **best** value between methods is highlighted for each object metric and mask state. Erosion slightly improves scores by focusing on core object geometry. Dilation degrades accuracy by including unmodified context. This shows our depth-based metric isolates removal-induced geometry changes and is robust to boundary noise.

| Scene | Object | Method | E 15 | E 10 | E 5 | Original | D 5 | D 10 | D 15 |
|-------|--------|--------|------|------|-----|----------|-----|------|------|
| Backyard | Deckchair | GC | **0.65** | 0.63 | **0.60** | **0.56** | **0.52** | 0.50 | 0.49 |
| | | AF | **0.65** | **0.64** | **0.60** | 0.54 | 0.51 | **0.51** | **0.51** |
| | Chairs | GC | **0.84** | **0.83** | **0.83** | **0.83** | **0.83** | **0.82** | **0.82** |
| | | AF | 0.67 | 0.66 | 0.65 | 0.62 | 0.61 | 0.60 | 0.58 |
| | Stroller | GC | **0.85** | **0.85** | **0.85** | **0.85** | **0.86** | **0.86** | **0.86** |
| | | AF | 0.72 | 0.72 | 0.72 | 0.72 | 0.72 | 0.72 | 0.73 |
| | Playhouse | GC | **0.50** | **0.50** | **0.50** | **0.50** | **0.50** | **0.50** | **0.50** |
| | | AF | 0.48 | 0.48 | 0.48 | 0.49 | 0.49 | 0.48 | 0.48 |
| | Toy Truck | GC | **0.21** | **0.21** | **0.22** | **0.22** | **0.22** | **0.23** | **0.23** |
| | | AF | 0.20 | 0.19 | 0.19 | 0.20 | 0.20 | 0.20 | 0.20 |
| Bedroom | Table | GC | **0.43** | **0.44** | **0.45** | **0.48** | **0.48** | **0.48** | **0.50** |
| | | AF | 0.38 | 0.39 | 0.40 | 0.44 | 0.45 | 0.45 | 0.46 |
| Living Room | Pillows | GC | **0.19** | **0.19** | **0.19** | **0.19** | **0.19** | **0.19** | **0.20** |
| | | AF | 0.18 | 0.18 | 0.18 | 0.18 | 0.18 | 0.18 | 0.19 |
| | Sofa | GC | **0.17** | **0.17** | **0.17** | **0.17** | **0.18** | **0.18** | **0.18** |
| | | AF | 0.13 | 0.13 | 0.13 | 0.13 | 0.13 | 0.13 | 0.13 |
| Office | Chairs | GC | **0.34** | **0.34** | **0.34** | **0.34** | **0.35** | **0.36** | **0.37** |
| | | AF | 0.32 | 0.32 | 0.33 | 0.33 | 0.33 | 0.32 | 0.32 |
| Park | Bicycle | GC | **0.68** | **0.68** | **0.68** | **0.68** | **0.68** | **0.68** | **0.68** |
| | | AF | 0.46 | 0.47 | 0.48 | 0.48 | 0.47 | 0.46 | 0.46 |
| Stairwell | Backpack | GC | **0.37** | **0.37** | **0.37** | **0.37** | **0.38** | **0.38** | **0.38** |
| | | AF | **0.37** | **0.37** | **0.37** | **0.37** | 0.37 | **0.38** | **0.38** |

Table 4: **sim$_{\text{SAM}}$ results for mask erosion (E) and dilation (D) analysis, Remove360 dataset.** Results show, how evaluation metric sim$_{\text{SAM}}$ changes when the ground truth mask is eroded or dilated by 5-15 pixels compared to the original mask used for Remove360 dataset. The **best** value between methods is highlighted for each object metric and mask state. Erosion sometimes improves scores (Deckchair: $0.56 \rightarrow 0.65$), while dilation has small, inconsistent effects. The metric is robust to mask variation, and variability across scenes highlights the need for multiple complementary metrics.

quality (Tab. 5, 6, 8, 9). The presence of redundancy in the metrics makes the proposed evaluation robust to potential errors in the semantic models used in the derivation.

When a method achieves good results on all three metrics (IOU$_{\text{drop}}$, acc$_{\Delta\text{depth}}$, sim$_{\text{SAM}}$), then it is very likely that the removal succeeded and that the metrics are reliable. However, a mix of good

and bad metric scores indicates that either the removal quality is low or that the segmentation used to derive $IOU_{drop}$ and $sim_{SAM}$ are incorrect, thus, the metric is not reliable. For instance, low drop in object detection IOUdrop can mean either a failed removal or poor segmentation output from the segmentation model. The other metrics help disambiguate between the two interpretations. Cross-checking with depth difference accuracy $acc_{\Delta depth}$ helps resolve this ambiguity. This metric tends to overestimate removal quality, making lower values a stronger indicator of failure—e.g., Aura Fusion Wu et al. (2025) on 'Pillows' (Tab. 5, Fig. 10a) However, a higher $acc_{\Delta depth}$ does not guarantee success. For example, in Fig. 9a is shown that Gaussian Cut Jain et al. (2024) achieves 0.62 $acc_{\Delta depth}$ on 'Sofa', but $IOU_{drop}$ is low (0.57). The third metric can disambiguate such a case: in this example of 'Sofa' removal, the $sim_{SAM}$ at 0.17 suggests that the removal does not perform well, as the SAM Kirillov et al. (2023) segmentation is not similar with ground truth SAM segmentation after removal. Therefore the high $acc_{\Delta depth}$ does indicate a high-quality removal but instead just some local editing, what we can visually confirm. Visual inspection confirms the local edits.

This analysis demonstrates the complementary nature of the metrics. Together, they provide robust, interpretable evaluation, especially in the absence of ground-truth post-removal data. This is critical on datasets like Mip-NeRF360, where only pre-removal ground truth is available; in such cases, $sim_{SAM}$ between before and after renders serves as a proxy, with lower values indicating better removal.

More quantitative and qualitative results follow in the next section.

## B  REMOVAL RESULTS FOR REMOVE360 DATASET

### B.1  QUANTITATIVE RESULTS AFTER REMOVAL

Tab. 5, 6 present quantitative results after removal on the Remove360 dataset. The removal methods Gaussian Cut Jain et al. (2024) and Aura Fusion Wu et al. (2025) perform relatively similar, with advantage of Gaussian Cut Jain et al. (2024) in the semantic similarity $sim_{SAM}$, and depth difference accuracy $acc_{\Delta depth}$. However, Aura Fusion Wu et al. (2025) performs better in the semantic object segmentation after removal, achieving less detections compared to Gaussian Cut Jain et al. (2024). Having ground-truth after removal, we are able to compute PSNR between the renders after removal and ground-truth novel views. Having visually consistent background after object removal is wanted. A higher PSNR means the removal method preserves the visual quality better. Slightly better results are achieved by Gaussian Cut Jain et al. (2024). Overall low PSNR values indicate poor quality after removal, and the need of additional processing, for example through in-painting (see Fig. 7c, 8a, 8b, 8c, 9b.

### B.2  QUALITATIVE RESULTS AFTER REMOVAL

Qualitative results for each scene in the Remove360 dataset are presented in Fig. 7, 8, 9, 10. Each visualization includes SAM Kirillov et al. (2023) segmentations, and depth differences computed before and after removal using the thresholding approach described in the spatial recognition subsection of the main paper. The segmentation similarity between the SAM segments of the ground-truth and the render after removal is reported; higher similarity scores indicate a closer match to the ground-truth and thus more successful object removal. Additionally, the accuracy of the depth difference within the ground-truth mask is reported; higher values suggest effective removal, as changes in depth at the object's location are expected. Gaussian Cut Jain et al. (2024) generally produces more visually coherent results, leaving fewer artifacts and preserving scene quality more effectively than competing methods. Among the methods, Gaussian Cut Jain et al. (2024) often produces the most visually coherent results, with fewer rendering artifacts and more realistic background compared to Aura Fusion360 Wu et al. (2025) results.

However, these visual results are not always aligned with the quantitative semantic metrics (see Tab.5,6). In some cases, even though the object appears to be successfully removed in the image (Fig. 6), the semantic segmentation metric ($IoU_{post}$) reports relatively high object detection after removal (office chairs after removal reach mean $IoU_{post}$ of 0.18 and 0.19, see Tab 6), indicating that semantic features of the removed object are still present. This discrepancy often points to invisible or occluded Gaussians that still carry semantic cues, which the SAM model can detect even when they're not visually obvious, see Fig. 6.

| Scene | Object | Method | IoU$_{\text{drop}}$ ↑ | acc$_{\text{seg, IoU}_{post} < 0.5}$ ↑ | acc$_{\Delta\text{depth}}$ ↑ | sim$_{\text{SAM}}$ ↑ | PSNR |
|---|---|---|---|---|---|---|---|
| Backyard | Deckchair | Gaussian Cut | **0.85** | **0.99** | **0.67** | **0.56** | 15.62 |
| | | Aura Fusion | 0.84 | **0.99** | 0.65 | 0.54 | **15.86** |
| | Chairs | Gaussian Cut | 0.85 | **1.00** | **0.76** | **0.83** | **17.99** |
| | | Aura Fusion | **0.87** | **1.00** | 0.67 | 0.62 | 17.89 |
| | Stroller | Gaussian Cut | **0.92** | **1.00** | **0.89** | **0.85** | **19.19** |
| | | Aura Fusion | 0.91 | **1.00** | 0.73 | 0.72 | 18.74 |
| | Playhouse | Gaussian Cut | 0.95 | **1.00** | **0.92** | **0.50** | 18.05 |
| | | Aura Fusion | **0.97** | **1.00** | 0.87 | 0.49 | **18.07** |
| | Toy Truck | Gaussian Cut | **0.95** | **0.99** | **0.73** | **0.22** | 15.62 |
| | | Aura Fusion | 0.93 | 0.98 | 0.64 | 0.20 | **15.66** |
| Bedroom | Table | Gaussian Cut | **0.91** | 0.98 | 0.57 | **0.48** | 21.76 |
| | | Aura Fusion | **0.91** | **1.00** | **0.58** | 0.44 | **21.92** |
| Living Room | Pillows | Gaussian Cut | 0.62 | 0.77 | **0.53** | **0.19** | **21.45** |
| | | Aura Fusion | **0.76** | **0.88** | 0.51 | 0.18 | 20.41 |
| | Sofa | Gaussian Cut | 0.57 | 0.50 | **0.62** | **0.17** | **17.45** |
| | | Aura Fusion | **0.62** | **0.64** | **0.62** | 0.13 | 16.81 |
| Office | Chairs | Gaussian Cut | **0.69** | **0.85** | **0.91** | **0.34** | **17.27** |
| | | Aura Fusion | 0.64 | 0.76 | 0.82 | 0.33 | 15.93 |
| Park | Bicycle | Gaussian Cut | **0.95** | 0.99 | **0.91** | **0.68** | **17.00** |
| | | Aura Fusion | **0.95** | **1.00** | 0.80 | 0.48 | 16.61 |
| Stairwell | Backpack | Gaussian Cut | **0.89** | **0.93** | **0.73** | **0.37** | **19.71** |
| | | Aura Fusion | 0.82 | 0.85 | 0.65 | **0.37** | 19.49 |

Table 5: **Remove360: Evaluation results.** These five metrics measure changes in semantics and depth before and after removal, along with image quality after removal: IoU$_{\text{drop}}$ measures the drop in semantic segmentation after removal, acc$_{\text{seg},\xi_{\text{IoU}}}$ measures the ratio of images after removal in which the semantic element is not recognized anymore while having IoU$_{\text{post}} < 0.5$. Values acc$_{\Delta\text{depth}}$ capture changes in the depth maps compared to object mask, sim$_{\text{SAM}}$ quantifies similarity in the SAM Kirillov et al. (2023) masks between ground truth and renders after removal, and PSNR measures of the whole image quality after removal, comparing the whole ground-truth novel view after removal to check for visual consistency. The **best** value is highlighted for each object and metric. GaussianCut (GC) Jain et al. (2024) outperforms AuraFusion (AF) Wu et al. (2025), especially in the instance segmentation similarity sim$_{\text{SAM}}$. Remaining methods, Gaussian Groupping Ye et al. (2024a), Feature3DGS Zhou et al. (2024) and SAGS Hu et al. (2024), were unable to run object removal, therefore they are not included in this table.

This highlights the importance of our grounded SAM-based metric, which serves as a proxy for semantic leakage: It can detect residual traces of the removed object that are not apparent in RGB renderings but remain in the underlying 3D representation. Thus, even if an image looks correct to a human observer, the scene may still reveal what was removed to a machine vision system—violating the goal of effective and irrecoverable object removal.

Our metric is particularly valuable in the absence of true post-removal ground-truth labels, as it leverages the consistency and sensitivity of a strong segmentation model to detect failures that would otherwise go unnoticed.

## B.3 ADDITIONAL ANALYSIS

We analyse the correlation between input view visibility and residual signal strength after object removal using the sim$_{\text{SAM}}$ metric. Visibility (IoU$_{\text{before}}$) is defined as the IoU between pre-removal semantic masks, obtained with GroundedSAM Ren et al. (2024) from rendered views, and ground truth masks from the original images. The sim$_{\text{SAM}}$ metric measures similarity between SAM segments of the removal ground truth image and the after-removal renderings, with higher values indicating better removal. For both Gaussian Cut (GC) and Aura Fusion (AF), are images grouped by IoU$_{\text{before}}$ ranges and computed the Pearson correlation coefficient $r$ between IoU$_{\text{before}}$ and sim$_{\text{SAM}}$. All results are displayed in Tab. 7. The hypothesis is that higher visibility should yield better removal, however the results yield that strong correlations are rare and mostly occurred in bins with very few samples (N

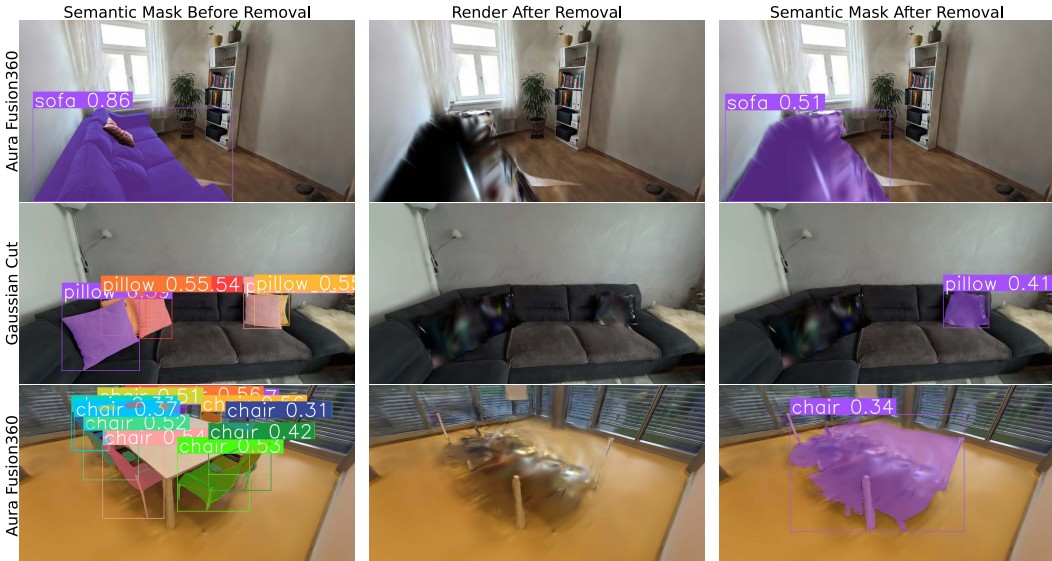

Figure 6: **Semantic segmentation changes before and after removal, Remove360 dataset.** Left-right: GroundedSAM2 Kirillov et al. (2023); Liu et al. (2023); Ren et al. (2024) overlay on the rendering before removal, rendering after removal, overlay after removal. These semantic masks are used to calculate change in semantic segmentation in $\text{IoU}_{\text{drop}}$ and its accuracy $\text{acc}_{\text{seg},\xi_{\text{IoU}}}$. Rows: Different object removals in dataset Remove360. Even though the object can not be recognized by a human, the segmentation model still finds it. One explanation can be that the pixel distribution on the edited area still exhibits patterns characteristic of the object, similar to what occurs in adversarial attacks.

¡ 20), limiting statistical reliability. A notable exception is Backyard Toy House, where for $\text{IoU}_{\text{before}}$ 0.95–1.00 both methods, GC ($r = –0.569$, N = 116) and AF ($r = –0.504$, N = 120), showed clear negative correlations, indicating that high visibility did not guarantee effective removal. In the highest-visibility ranges (0.90–1.00), covering most of the dataset, correlations were generally weak (–0.2 to +0.2) and inconsistent across scenes and methods. These results suggest that per-image visibility alone is not a reliable predictor of removal quality, and residual behaviour is likely influenced by more complex multi-view factors.

## C  REMOVAL RESULTS FOR MIP-NERF360 DATASET BARRON ET AL. (2022)

### C.1  QUANTITATIVE RESULTS AFTER REMOVAL

Tab. 8, 9 present quantitative results after removal on the Mip-NERF360 dataset. The removal methods Gaussian Cut Jain et al. (2024) and Aura Fusion Wu et al. (2025) achieve the best performance in reducing semantic segmentation and object detection presence. However, some objects remain difficult to fully remove, such as 'Slippers' (see Fig. 11a) and 'Blue Gloves' (see Fig. 11b), which are still detected in up to 96% of the views, with at least 11% persistence across methods. Note that both methods were designed and evaluated on Mip-NeRF360 Barron et al. (2022). We don't know whether the segmentation model used in our evaluation, GroundedSAM2 Kirillov et al. (2023); Liu et al. (2023); Ren et al. (2024), have been trained using this dataset. We were unable to obtain confirmation either confirming or denying this possibility. For this reason, results on the novel Remove360 dataset are considered more reliable. Since ground-truth novel views after removal are not available, PSNR cannot be computed for this dataset.

## C.2 QUALITATIVE RESULTS AFTER REMOVAL

Qualitative results for each used scene in the Mip-NERF360 dataset are presented in Fig. 13, 14, 15. Each visualization includes SAM Kirillov et al. (2023) segmentations, and depth differences computed before and after removal using a thresholding approach described in the spatial recognition subsection of the main paper. The semantic similarity between the SAM segments of renders before and after removal is reported; as ground-truth segmentation after removal is not available. Lower similarity scores indicate greater distinction between the before and after states, reflecting more successful object removal. Additionally, the accuracy of the depth difference within the pseudo-ground-truth mask is shown; higher values suggest effective removal, as changes in depth at the object's location are expected.

Gaussian Cut Jain et al. (2024) generally produces more visually coherent results, leaving fewer artifacts and preserving scene quality more effectively than competing methods. These observations are not supported by all the quantitative results of the semantic segmentation (Tab. 5, 6), which means invisible Gaussians with the semantic information, must be still present in the image, see Fig. 12.

| Scene | Object | Method | $\text{mIoU}_\text{pre}$ | $\text{mIoU}_\text{post}$ | $\text{IoU}_\text{drop}$ ↑ |
|---|---|---|---|---|---|
| Backyard | Deckchair | Gaussian Cut | 0.90 | 0.05 | **0.85** |
| | | Aura Fusion | 0.88 | 0.04 | 0.84 |
| | Chairs | Gaussian Cut | 0.89 | 0.04 | 0.85 |
| | | Aura Fusion | 0.89 | 0.02 | **0.87** |
| | Stroller | Gaussian Cut | 0.92 | 0.00 | **0.92** |
| | | Aura Fusion | 0.91 | 0.00 | 0.91 |
| | Playhouse | Gaussian Cut | 0.97 | 0.03 | 0.95 |
| | | Aura Fusion | 0.99 | 0.02 | **0.97** |
| | Toy Truck | Gaussian Cut | 0.95 | 0.00 | **0.95** |
| | | Aura Fusion | 0.95 | 0.02 | 0.93 |
| Bedroom | Table | Gaussian Cut | 0.93 | 0.02 | **0.91** |
| | | Aura Fusion | 0.92 | 0.01 | **0.91** |
| Living Room | Pillows | Gaussian Cut | 0.90 | 0.28 | 0.62 |
| | | Aura Fusion | 0.89 | 0.13 | **0.76** |
| | Sofa | Gaussian Cut | 0.96 | 0.39 | 0.57 |
| | | Aura Fusion | 0.95 | 0.33 | **0.62** |
| Office | Chairs | Gaussian Cut | 0.85 | 0.18 | **0.67** |
| | | Aura Fusion | 0.83 | 0.19 | 0.64 |
| Park | Bicycle | Gaussian Cut | 0.97 | 0.02 | **0.95** |
| | | Aura Fusion | 0.95 | 0.00 | **0.95** |
| Stairwell | Backpack | Gaussian Cut | 0.94 | 0.05 | **0.89** |
| | | Aura Fusion | 0.96 | 0.14 | 0.82 |

(a) **Breakdown of the proposed semantic segmentation $\text{IoU}_\text{drop}$ metric.** $\text{IoU}_\text{drop} = \text{IoU}_\text{post}$ - $\text{IoU}_\text{pre}$ and the higher, the better the removal. The best-performing method per object is highlighted in bold. The mean individual segmentation IoUs before and after removal, $\text{mIoU}_\text{pre}$ and $\text{mIoU}_\text{post}$ respectively, are also reported.

| Scene | Object | Method | $\text{acc}_{\text{IoU}_\text{post} < 0.3}$ ↑ | $\text{acc}_{\text{IoU}_\text{post} < 0.5}$ ↑ | $\text{acc}_{\text{IoU}_\text{post} < 0.7}$ ↑ | $\text{acc}_{\text{IoU}_\text{post} < 0.9}$ ↑ |
|---|---|---|---|---|---|---|
| Backyard | Deckchair | Gaussian Cut | 0.903 | 0.987 | 0.992 | 0.992 |
| | | Aura Fusion | 0.932 | 0.987 | 0.992 | 1.000 |
| | Chairs | Gaussian Cut | 0.990 | 1.000 | 1.000 | 1.000 |
| | | Aura Fusion | 0.990 | 1.000 | 1.000 | 1.000 |
| | Stroller | Gaussian Cut | 1.000 | 1.000 | 1.000 | 1.000 |
| | | Aura Fusion | 1.000 | 1.000 | 1.000 | 1.000 |
| | Playhouse | Gaussian Cut | 0.980 | 1.000 | 1.000 | 1.000 |
| | | Aura Fusion | 0.995 | 1.000 | 1.000 | 1.000 |
| | Toy Truck | Gaussian Cut | 0.995 | 0.995 | 1.000 | 1.000 |
| | | Aura Fusion | 0.962 | 0.978 | 0.984 | 0.984 |
| Bedroom | Table | Gaussian Cut | 0.973 | 0.980 | 0.993 | 0.993 |
| | | Aura Fusion | 1.000 | 1.000 | 1.000 | 1.000 |
| Living Room | Pillows | Gaussian Cut | 0.738 | 0.767 | 0.865 | 0.877 |
| | | Aura Fusion | 0.877 | 0.883 | 0.890 | 0.914 |
| | Sofa | Gaussian Cut | 0.483 | 0.500 | 0.614 | 0.977 |
| | | Aura Fusion | 0.625 | 0.642 | 0.676 | 0.795 |
| Office | Chairs | Gaussian Cut | 0.793 | 0.845 | 0.942 | 0.997 |
| | | Aura Fusion | 0.735 | 0.761 | 0.851 | 0.977 |
| Park | Bicycle | Gaussian Cut | 0.990 | 0.995 | 1.000 | 1.000 |
| | | Aura Fusion | 1.000 | 1.000 | 1.000 | 1.000 |
| Stairwell | Backpack | Gaussian Cut | 0.904 | 0.930 | 0.995 | 1.000 |
| | | Aura Fusion | 0.727 | 0.850 | 0.989 | 1.000 |

(b) **Breakdown of semantic recognition accuracy $\text{acc}_{\text{seg},\xi_\text{IoU}}$ by $\text{IoU}_\text{post}$ threshold.** This table shows the percentage of images where the object is no longer recognized, using IoU thresholds $\{0.3, 0.5, 0.7, 0.9\}$ to define recognition. Higher values indicate better removal. Both methods succeed in removing semantics in over 90% of cases, except for objects like 'Sofa', where over 50% of images retain ¿30% IoU overlap. Similarly, 'Pillows' and 'Chairs' retain semantics in ~75% of cases at the 0.3 threshold. For visual results see Fig. 6

Table 6: **Remove360: Additional evaluation of the object segmentation after removal.**

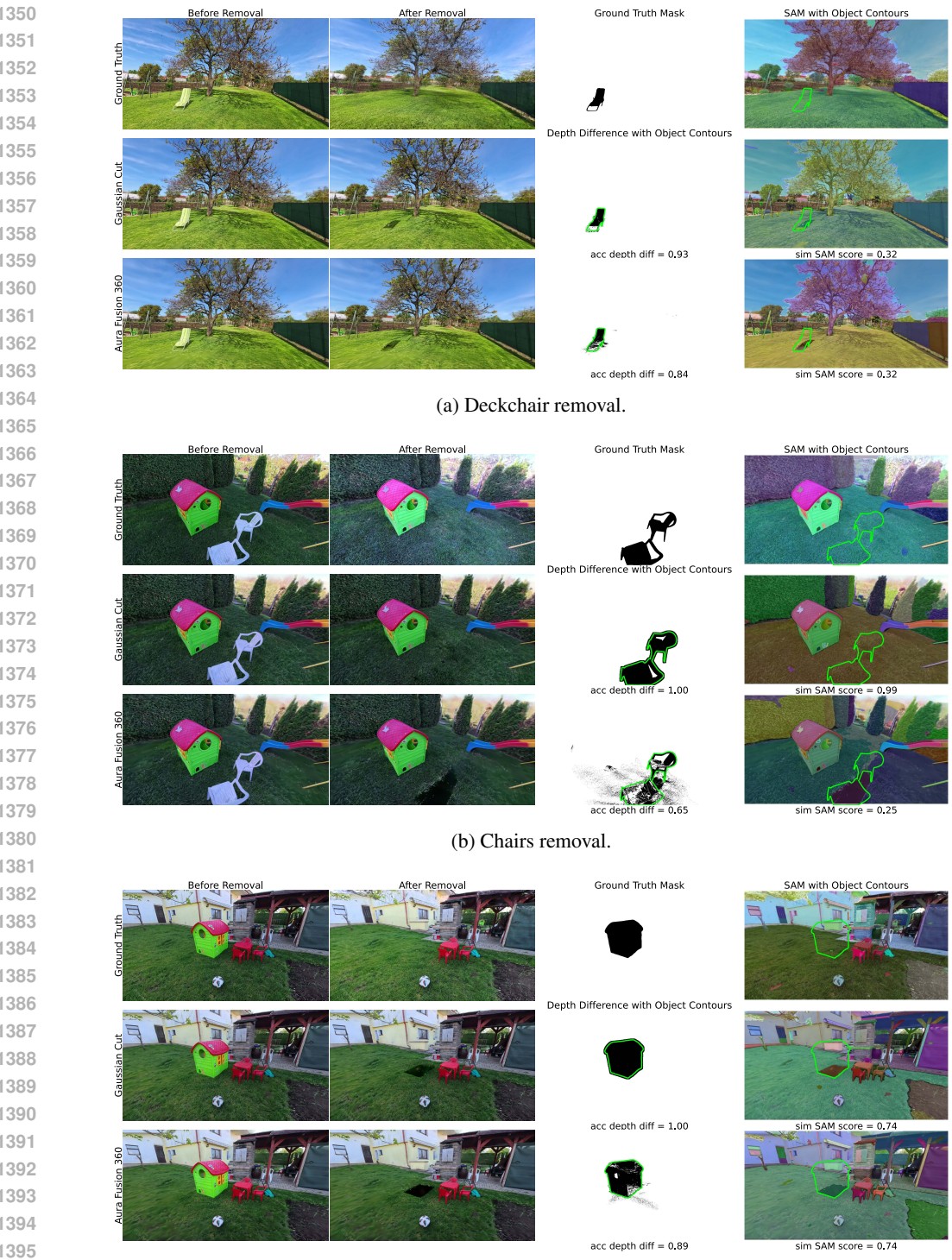

(a) Deckchair removal.

(b) Chairs removal.

(c) Playhouse removal.

Figure 7: **Remove360: Visual comparison of object removal results.** Each row shows results for: ground-truth (top), Gaussian Cut (GC) Jain et al. (2024) (middle), and Aura Fusion Wu et al. (2025) (bottom). Each triplet displays: before removal, result after removal, and evaluation (either ground-truth mask or depth difference with mask accuracy, and SAM Kirillov et al. (2023) masks with similarity to the ground-truth). Higher depth difference accuracy and higher SAM similarity score suggest better removal. GC often achieves more consistent background reconstruction, particularly visible in comparison to ground-truth novel views.

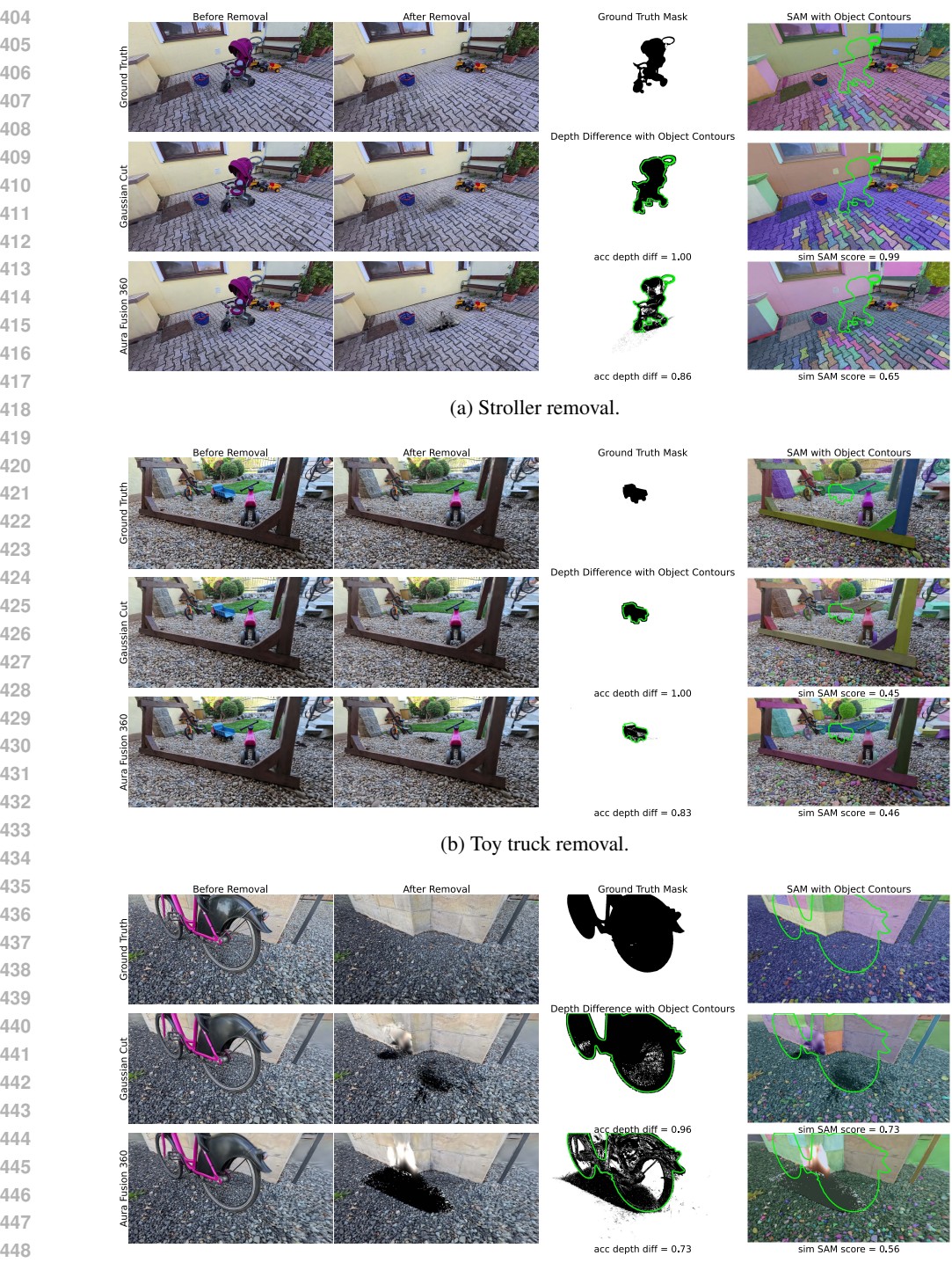

(a) Stroller removal.

(b) Toy truck removal.

(c) Bicycle removal.

Figure 8: **Remove360: Visual comparison of object removal results.** Each row shows results for: ground-truth (top), Gaussian Cut (GC) Jain et al. (2024) (middle), and Aura Fusion Wu et al. (2025) (bottom). Each triplet displays: before removal, result after removal, and evaluation (either ground-truth mask or depth difference with mask accuracy, and SAM Kirillov et al. (2023) masks with similarity to the ground-truth). Higher depth difference accuracy and higher SAM similarity score suggest better removal. GC often achieves more consistent background reconstruction, particularly visible in comparison to ground-truth novel views.

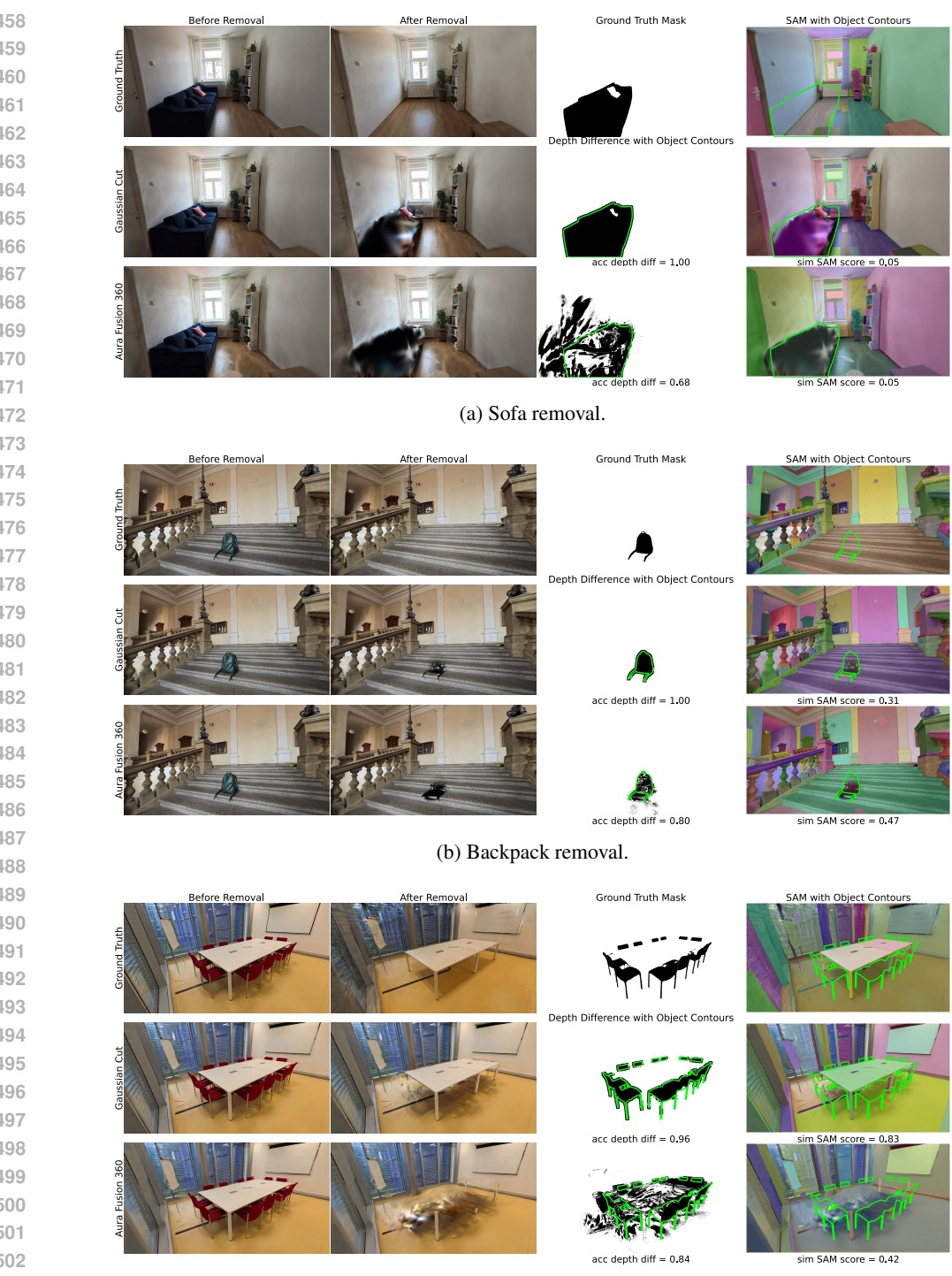

Figure 9: **Remove360: Visual comparison of object removal results.** Each row shows results for: ground-truth (top), Gaussian Cut (GC) Jain et al. (2024) (middle), and Aura Fusion Wu et al. (2025) (bottom). Each triplet displays: before removal, result after removal, and evaluation (either ground-truth mask or depth difference with mask accuracy, and SAM Kirillov et al. (2023) masks with similarity to the ground-truth). Higher depth difference accuracy and higher SAM similarity score suggest better removal. GC often achieves more consistent background reconstruction, particularly visible in comparison to ground-truth novel views.

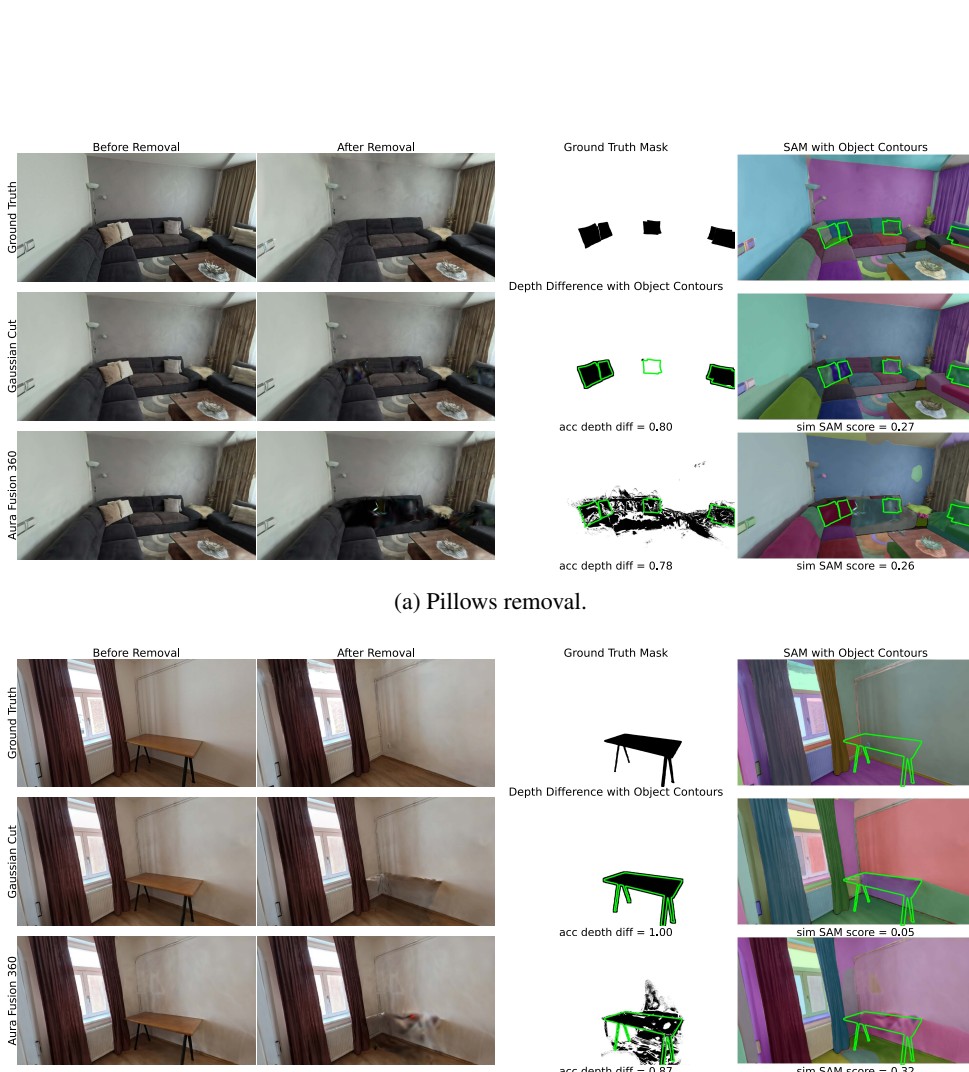

Figure 10: **Remove360: Visual comparison of object removal results.** Each row shows results for: ground-truth (top), Gaussian Cut Jain et al. (2024) (middle), and Aura Fusion Wu et al. (2025) (bottom) method. Each triplet displays: before removal, result after removal, and evaluation (either ground-truth mask or depth difference with mask accuracy, and SAM Kirillov et al. (2023) masks with similarity to the ground-truth). Higher depth difference accuracy and higher SAM similarity score suggest better removal. Gaussian Cut often achieves more consistent background reconstruction, particularly visible in comparison to ground-truth novel views.

| Scene | Object | Method | 0.01–0.50 | | 0.50–0.75 | | 0.75–0.90 | | 0.90–0.95 | | 0.95–1.00 | |
|---|---|---|---|---|---|---|---|---|---|---|---|---|
| | | | #img | $r$ | #img | $r$ | #img | $r$ | #img | $r$ | #img | $r$ |
| Backyard | Deckchair | GC | 1 | – | 21 | -0.215 | 68 | -0.007 | 50 | -0.012 | 83 | -0.099 |
| | | AF | 1 | – | 21 | -0.225 | 68 | -0.001 | 50 | -0.010 | 83 | -0.088 |
| | White chairs | GC | 11 | -0.465 | 1 | – | 2 | 1.000 | 9 | -0.403 | 115 | 0.120 |
| | | AF | 17 | -0.196 | 1 | – | 3 | 0.883 | 8 | 0.163 | 152 | 0.054 |
| | Stroller | GC | 2 | – | 0 | – | 1 | – | 5 | 0.579 | 193 | 0.061 |
| | | AF | 3 | – | 0 | – | 2 | 1.000 | 10 | -0.112 | 186 | 0.091 |
| | Toy house | GC | 4 | – | 0 | – | 1 | – | 0 | – | 116 | -0.569 |
| | | AF | 0 | – | 0 | – | 1 | – | 0 | – | 120 | -0.504 |
| | Toy truck | GC | 6 | 0.215 | 1 | – | 2 | – | 1 | – | 172 | -0.136 |
| | | AF | 6 | 0.305 | 0 | – | 2 | 1.000 | 4 | -0.979 | 170 | 0.012 |
| Bedroom | Table | GC | 6 | 0.008 | 0 | – | 4 | -0.645 | 18 | -0.117 | 119 | 0.020 |
| | | AF | 7 | 0.488 | 0 | – | 6 | -0.054 | 34 | -0.082 | 100 | 0.076 |
| Living room | Pillows | GC | 14 | -0.145 | 6 | -0.239 | 5 | 0.347 | 4 | -0.970 | 132 | -0.018 |
| | | AF | 15 | -0.111 | 8 | 0.440 | 3 | -0.995 | 7 | 0.161 | 128 | -0.182 |
| Office | Chairs | GC | 9 | -0.134 | 33 | 0.071 | 123 | 0.211 | 121 | 0.131 | 55 | -0.170 |
| | | AF | 12 | -0.143 | 32 | 0.015 | 197 | -0.002 | 92 | -0.168 | 8 | 0.214 |
| Park | Bicycle | GC | 0 | – | 0 | – | 17 | 0.218 | 16 | -0.523 | 149 | -0.031 |
| | | AF | 0 | – | 5 | -0.098 | 26 | 0.184 | 15 | -0.365 | 136 | 0.066 |
| Stairwell | Backpack | GC | 2 | – | 0 | – | 0 | – | 5 | 0.298 | 102 | -0.177 |
| | | AF | 0 | – | 0 | – | 4 | 0.441 | 6 | -0.156 | 106 | -0.120 |

Table 7: **Correlation analysis between object visibility ranges before removal and $sim_{SAM}$ score after removal on the Remove360 dataset.** Pearson correlation statistic noted as $r$. Positive $r$: higher visibility $\rightarrow$ higher $sim_{SAM}$. Negative $r$: higher visibility $\rightarrow$ lower $sim_{SAM}$. Close to 0: little or no correlation. Notation "–" means insufficient data. Weak and inconsistent correlations indicate that visibility per image alone is not a reliable predictor of removal quality. Strong correlations mostly occur in $IoU_{before}$ bins with very few samples (N ¡ 20, 10% of total views), limiting statistical reliability. The image distribution is skewed toward the high IoU ranges. This is not surprising, because the benchmark is designed to have high object visibility in most views, leaving only few low-IoU views.

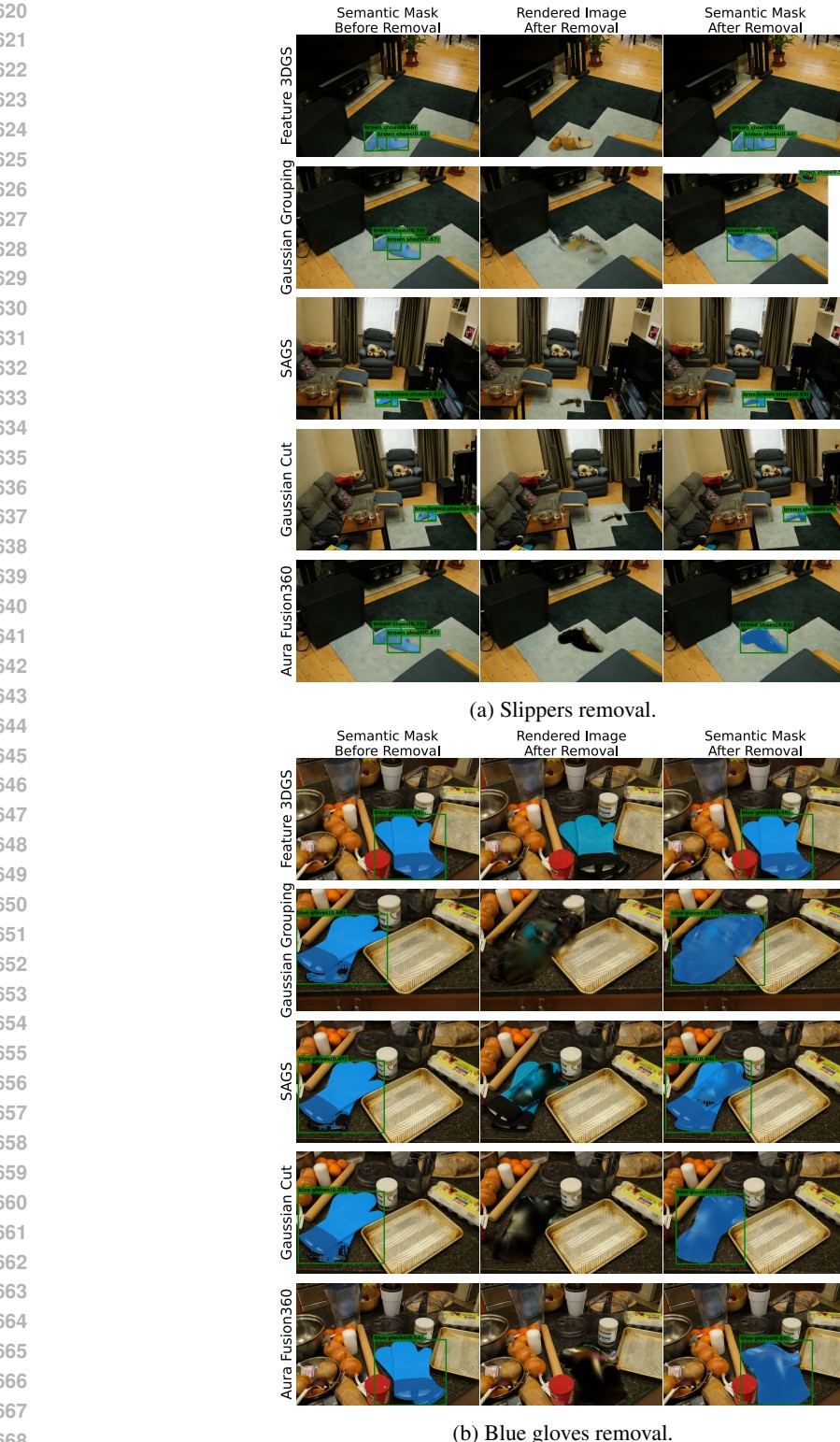

(a) Slippers removal.

(b) Blue gloves removal.

Figure 11: **Semantic segmentation changes before and after object removal, Mip-NERF360 Barron et al. (2022) dataset.** Left-right: GroundedSAM2 Kirillov et al. (2023); Liu et al. (2023); Ren et al. (2024) overlay on the rendering before removal, rendering after removal, overlay after removal. These semantic masks are used to calculate change in semantic segmentation in $IoU_{drop}$ and its accuracy $acc_{seg, \xi_{IoU}}$. Rows: Different methods applied on different objects. Even though the object can not be recognized by a human, the segmentation model finds it.

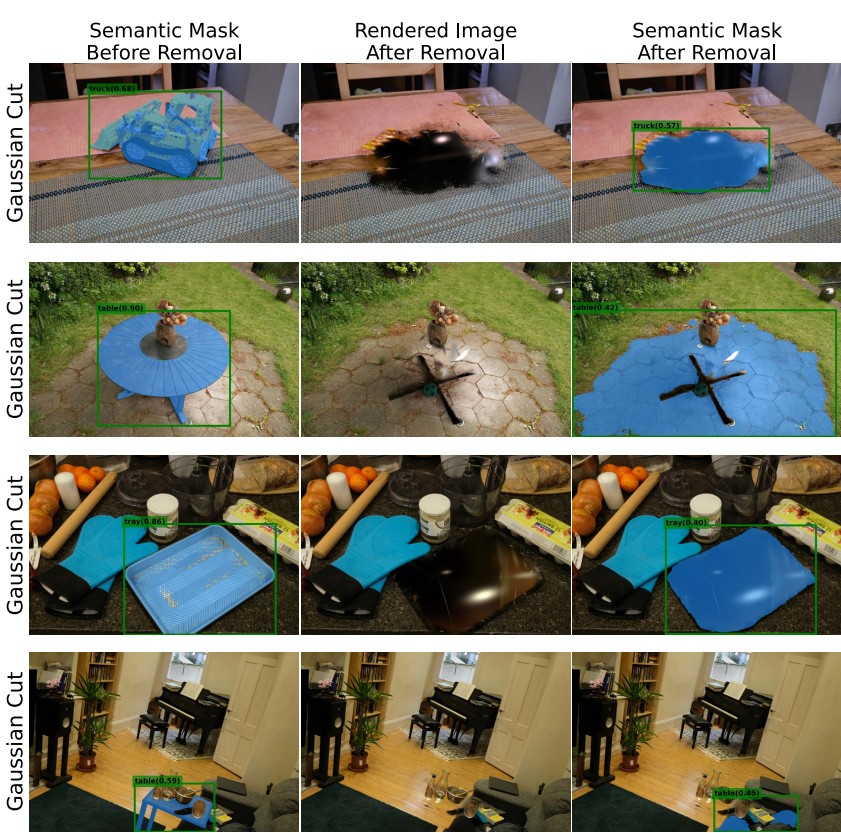

Figure 12: **Semantic segmentation changes before and after object removal of Gaussian Cut Jain et al. (2024) method, Mip-NERF360 Barron et al. (2022) dataset.** Left-right: GroundedSAM2 Kirillov et al. (2023); Liu et al. (2023); Ren et al. (2024) overlay on the rendering before removal, rendering after removal, overlay after removal. These semantic masks are used to calculate change in semantic segmentation in $\text{IoU}_{\text{drop}}$ and its accuracy $\text{acc}_{\text{seg},\xi_{\text{IoU}}}$. Rows: Different objects removed by Gaussian Cut Jain et al. (2024) in different scenes from Mip-NERF360 Barron et al. (2022). Even though the object can not be recognized by a human, the segmentation model still finds it.

| Scene | Object | Method | mIoU$_{pre}$ | mIoU$_{post}$ | IoU$_{drop}$ ↑ |
|---|---|---|---|---|---|
| Counter | Baking Tray | Gaussian Grouping | 0.61 | 0.08 | 0.53 |
| | | Feature 3DGS | 0.54 | 0.21 | 0.34 |
| | | SAGS | 0.62 | 0.52 | 0.10 |
| | | Gaussian Cut | 0.63 | 0.01 | **0.62** |
| | | Aura Fusion | 0.64 | 0.04 | 0.60 |
| | Plant | Gaussian Grouping | 0.84 | 0.00 | 0.84 |
| | | Feature 3DGS | 0.75 | 0.00 | 0.75 |
| | | SAGS | 0.85 | 0.82 | 0.03 |
| | | Gaussian Cut | 0.86 | 0.00 | 0.86 |
| | | Aura Fusion | 0.87 | 0.00 | **0.87** |
| | Blue Gloves | Gaussian Grouping | 0.75 | 0.15 | 0.60 |
| | | Feature 3DGS | 0.67 | 0.66 | 0.01 |
| | | SAGS | 0.74 | 0.64 | 0.10 |
| | | Gaussian Cut | 0.74 | 0.15 | 0.60 |
| | | Aura Fusion | 0.76 | 0.11 | **0.65** |
| | Egg Box | Gaussian Grouping | 0.63 | 0.00 | **0.63** |
| | | Feature 3DGS | 0.78 | 0.70 | 0.08 |
| | | SAGS | 0.60 | 0.04 | 0.56 |
| | | Gaussian Cut | 0.63 | 0.01 | 0.62 |
| | | Aura Fusion | 0.64 | 0.01 | **0.63** |
| Room | Plant | Gaussian Grouping | 0.50 | 0.23 | 0.26 |
| | | Feature 3DGS | 0.53 | 0.00 | **0.53** |
| | | SAGS | 0.52 | 0.35 | 0.17 |
| | | Gaussian Cut | 0.53 | 0.00 | **0.53** |
| | | Aura Fusion | 0.49 | 0.26 | 0.23 |
| | Slippers | Gaussian Grouping | 0.96 | 0.14 | **0.82** |
| | | Feature 3DGS | 0.96 | 0.96 | 0.00 |
| | | SAGS | 0.96 | 0.71 | 0.25 |
| | | Gaussian Cut | 0.97 | 0.48 | 0.48 |
| | | Aura Fusion | 0.43 | 0.37 | 0.06 |
| | Coffee Table | Gaussian Grouping | 0.88 | 0.02 | **0.86** |
| | | Feature 3DGS | 0.86 | 0.29 | 0.57 |
| | | SAGS | 0.89 | 0.89 | 0.00 |
| | | Gaussian Cut | 0.89 | 0.03 | **0.86** |
| | | Aura Fusion | 0.58 | 0.03 | 0.55 |
| Kitchen | Truck | Gaussian Grouping | 0.67 | 0.06 | 0.61 |
| | | Feature 3DGS | 0.67 | 0.05 | 0.62 |
| | | SAGS | 0.67 | 0.00 | 0.67 |
| | | Gaussian Cut | 0.67 | 0.01 | 0.66 |
| | | Aura Fusion | 0.96 | 0.01 | **0.95** |
| Garden | Table | Gaussian Grouping | 0.89 | 0.41 | 0.48 |
| | | Feature 3DGS | 0.90 | 0.24 | 0.67 |
| | | SAGS | 0.90 | 0.09 | 0.81 |
| | | Gaussian Cut | 0.90 | 0.04 | 0.86 |
| | | Aura Fusion | 0.91 | 0.01 | **0.90** |
| | Ball | Gaussian Grouping | 0.16 | 0.00 | 0.16 |
| | | Feature 3DGS | 0.06 | 0.06 | 0.00 |
| | | SAGS | 0.41 | 0.00 | 0.41 |
| | | Gaussian Cut | 0.42 | 0.00 | **0.42** |
| | | Aura Fusion | 0.42 | 0.00 | **0.42** |
| | Vase | Gaussian Grouping | 0.85 | 0.22 | 0.64 |
| | | Feature 3DGS | 0.90 | 0.11 | 0.79 |
| | | SAGS | 0.97 | 0.01 | 0.96 |
| | | Gaussian Cut | 0.97 | 0.01 | **0.97** |
| | | Aura Fusion | 0.98 | 0.01 | **0.97** |

Table 8: **Mip-NERF360: Breakdown of the proposed semantic segmentation IoU$_{drop}$ metric.** IoU$_{drop}$ = IoU$_{post}$ - IoU$_{pre}$ and the higher, the better the removal. The best-performing method is highlighted in bold, second-best underlined. The mean individual segmentation IoUs before and after removal, mIoU$_{pre}$ and mIoU$_{post}$ respectively, are also reported.

| Scene | Object | Method | $\mathrm{acc}_{\mathrm{IoU}_{post} < 0.3}$ ↑ | $\mathrm{acc}_{\mathrm{IoU}_{post} < 0.5}$ ↑ | $\mathrm{acc}_{\mathrm{IoU}_{post} < 0.7}$ ↑ | $\mathrm{acc}_{\mathrm{IoU}_{post} < 0.9}$ ↑ |
|---|---|---|---|---|---|---|
| Counter | Baking Tray | Gaussian Grouping | 0.915 | 0.915 | 0.915 | 0.943 |
| | | Feature 3DGS | 0.783 | 0.783 | 0.802 | 0.821 |
| | | SAGS | 0.358 | 0.481 | 0.557 | 0.698 |
| | | Gaussian Cut | 0.991 | 0.991 | 0.991 | 0.991 |
| | | Aura Fusion | 0.953 | 0.953 | 0.953 | 0.962 |
| | Plant | Gaussian Grouping | 1.000 | 1.000 | 1.000 | 1.000 |
| | | Feature 3DGS | 1.000 | 1.000 | 1.000 | 1.000 |
| | | SAGS | 0.167 | 0.167 | 0.167 | 0.167 |
| | | Gaussian Cut | 1.000 | 1.000 | 1.000 | 1.000 |
| | | Aura Fusion | 1.000 | 1.000 | 1.000 | 1.000 |
| | Blue Gloves | Gaussian Grouping | 0.837 | 0.837 | 0.837 | 0.904 |
| | | Feature 3DGS | 0.240 | 0.279 | 0.337 | 0.471 |
| | | SAGS | 0.212 | 0.337 | 0.500 | 0.558 |
| | | Gaussian Cut | 0.827 | 0.827 | 0.827 | 0.962 |
| | | Aura Fusion | 0.875 | 0.885 | 0.885 | 0.962 |
| | Egg Box | Gaussian Grouping | 1.000 | 1.000 | 1.000 | 1.000 |
| | | Feature 3DGS | 0.196 | 0.196 | 0.206 | 0.289 |
| | | SAGS | 0.959 | 0.959 | 0.959 | 0.959 |
| | | Gaussian Cut | 0.990 | 0.990 | 1.000 | 1.000 |
| | | Aura Fusion | 0.990 | 0.990 | 0.990 | 0.990 |
| Room | Plant | Gaussian Grouping | 0.640 | 0.800 | 0.880 | 0.920 |
| | | Feature 3DGS | 1.000 | 1.000 | 1.000 | 1.000 |
| | | SAGS | 0.440 | 0.720 | 0.800 | 0.920 |
| | | Gaussian Cut | 1.000 | 1.000 | 1.000 | 1.000 |
| | | Aura Fusion | 0.961 | 0.961 | 0.981 | 0.994 |
| | Slippers | Gaussian Grouping | 0.853 | 0.853 | 0.853 | 0.868 |
| | | Feature 3DGS | 0.015 | 0.015 | 0.015 | 0.015 |
| | | SAGS | 0.029 | 0.279 | 0.324 | 0.779 |
| | | Gaussian Cut | 0.338 | 0.441 | 0.588 | 0.838 |
| | | Aura Fusion | 0.568 | 0.568 | 0.574 | 0.839 |
| | Coffee Table | Gaussian Grouping | 0.990 | 0.990 | 0.990 | 0.990 |
| | | Feature 3DGS | 0.586 | 0.616 | 0.788 | 0.949 |
| | | SAGS | 0.091 | 0.091 | 0.101 | 0.101 |
| | | Gaussian Cut | 0.970 | 0.990 | 0.990 | 0.990 |
| | | Aura Fusion | 0.981 | 0.961 | 0.981 | 0.994 |
| Kitchen | Truck | Gaussian Grouping | 0.897 | 0.922 | 0.990 | 1.000 |
| | | Feature 3DGS | 0.941 | 0.951 | 0.951 | 0.956 |
| | | SAGS | 1.000 | 1.000 | 1.000 | 1.000 |
| | | Gaussian Cut | 0.985 | 0.995 | 1.000 | 1.000 |
| | | Aura Fusion | 0.993 | 1.000 | 1.000 | 1.000 |
| Garden | Table | Gaussian Grouping | 0.426 | 0.541 | 0.709 | 0.899 |
| | | Feature 3DGS | 0.676 | 0.703 | 0.818 | 1.000 |
| | | SAGS | 0.872 | 0.878 | 0.926 | 1.000 |
| | | Gaussian Cut | 0.953 | 0.953 | 0.966 | 1.000 |
| | | Aura Fusion | 1.000 | 1.000 | 1.000 | 1.000 |
| | Ball | Gaussian Grouping | 1.000 | 1.000 | 1.000 | 1.000 |
| | | Feature 3DGS | 0.939 | 0.939 | 0.939 | 0.939 |
| | | SAGS | 1.000 | 1.000 | 1.000 | 1.000 |
| | | Gaussian Cut | 1.000 | 1.000 | 1.000 | 1.000 |
| | | Aura Fusion | 1.000 | 1.000 | 1.000 | 1.000 |
| | Vase | Gaussian Grouping | 0.784 | 0.784 | 0.797 | 0.804 |
| | | Feature 3DGS | 0.885 | 0.892 | 0.905 | 0.926 |
| | | SAGS | 1.000 | 1.000 | 1.000 | 1.000 |
| | | Gaussian Cut | 1.000 | 1.000 | 1.000 | 1.000 |
| | | Aura Fusion | 1.000 | 1.000 | 1.000 | 1.000 |

Table 9: **Mip-NERF360: Breakdown of the proposed metric of semantic recognition $\mathrm{acc}_{\mathrm{seg},\xi_{\mathrm{IoU}}}$ based on the $\mathrm{IoU}_{post}$ threshold.** This table presents the ratio of images in which the semantic element is not recognized. We define that the object is not segmented if the semantic segmentation IoU is lower than a threshold. The higher, the better the removal. Reported thresholds $\{0.3, 0.5, 0.7, 0.9\}$.

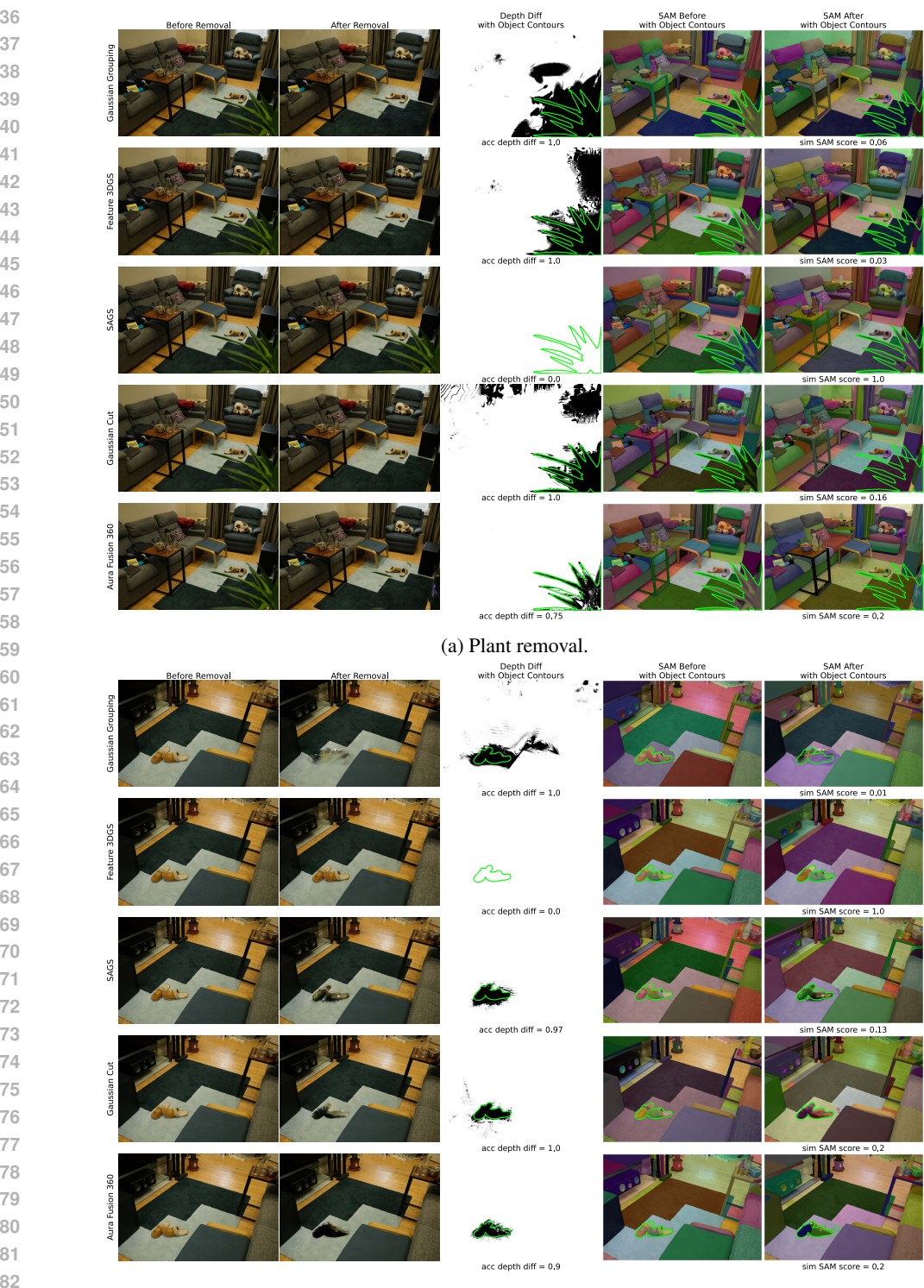

(a) Plant removal.

(b) Slippers removal.

Figure 13: **MipNERF360: Visual comparison of object removal results.** Each row shows results from Gaussian Grouping (GG) Ye et al. (2024a), Feature 3DGS Zhou et al. (2024), SAGS Hu et al. (2024), Gaussian Cut (GC) Jain et al. (2024), and Aura Fusion Wu et al. (2025). Each triplet includes before removal render, removal result, and evaluation—depth difference accuracy and SAM Kirillov et al. (2023) similarity to the input. Higher accuracy and lower similarity indicate better removal. GC Landrieu & Obozinski (2017) excels at removing plants; GG Ye et al. (2024a) performs best on slippers.

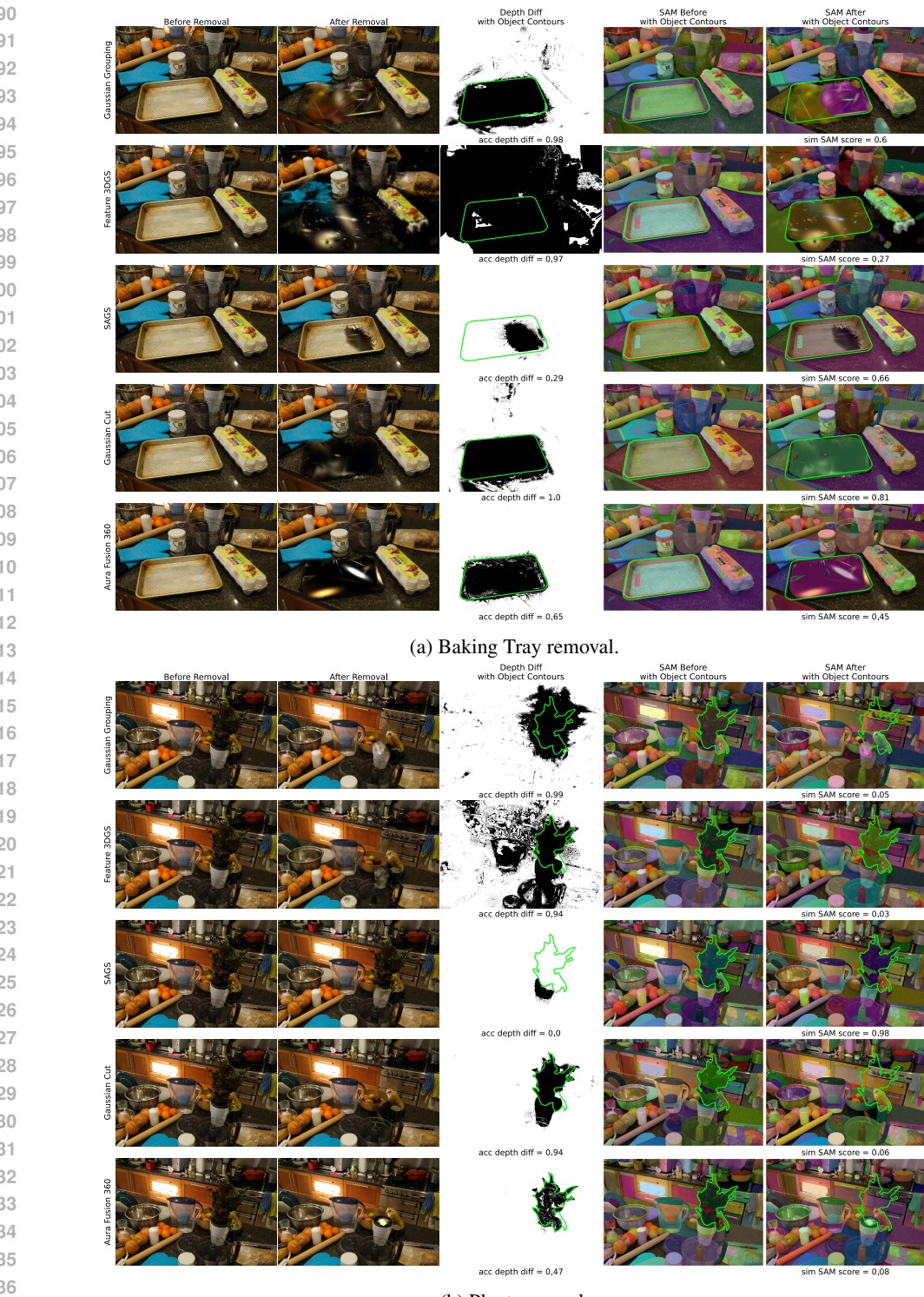

(a) Baking Tray removal.

(b) Plant removal.

Figure 14: **MipNERF360: Visual comparison of object removal results.** Each row shows results from Gaussian Grouping (GG) Ye et al. (2024a), Feature 3DGS Zhou et al. (2024), SAGS Hu et al. (2024), Gaussian Cut (GC) Jain et al. (2024), and Aura Fusion Wu et al. (2025). Each triplet includes the before removal render, removal result, and evaluation: depth difference accuracy within the object mask, and SAM Kirillov et al. (2023) similarity to the input. Higher depth accuracy and lower SAM similarity suggest better removal. Performance varies by object; GG Ye et al. (2024a) and GC Landrieu & Obozinski (2017) are best for plants, while the baking tray has no clear winner.

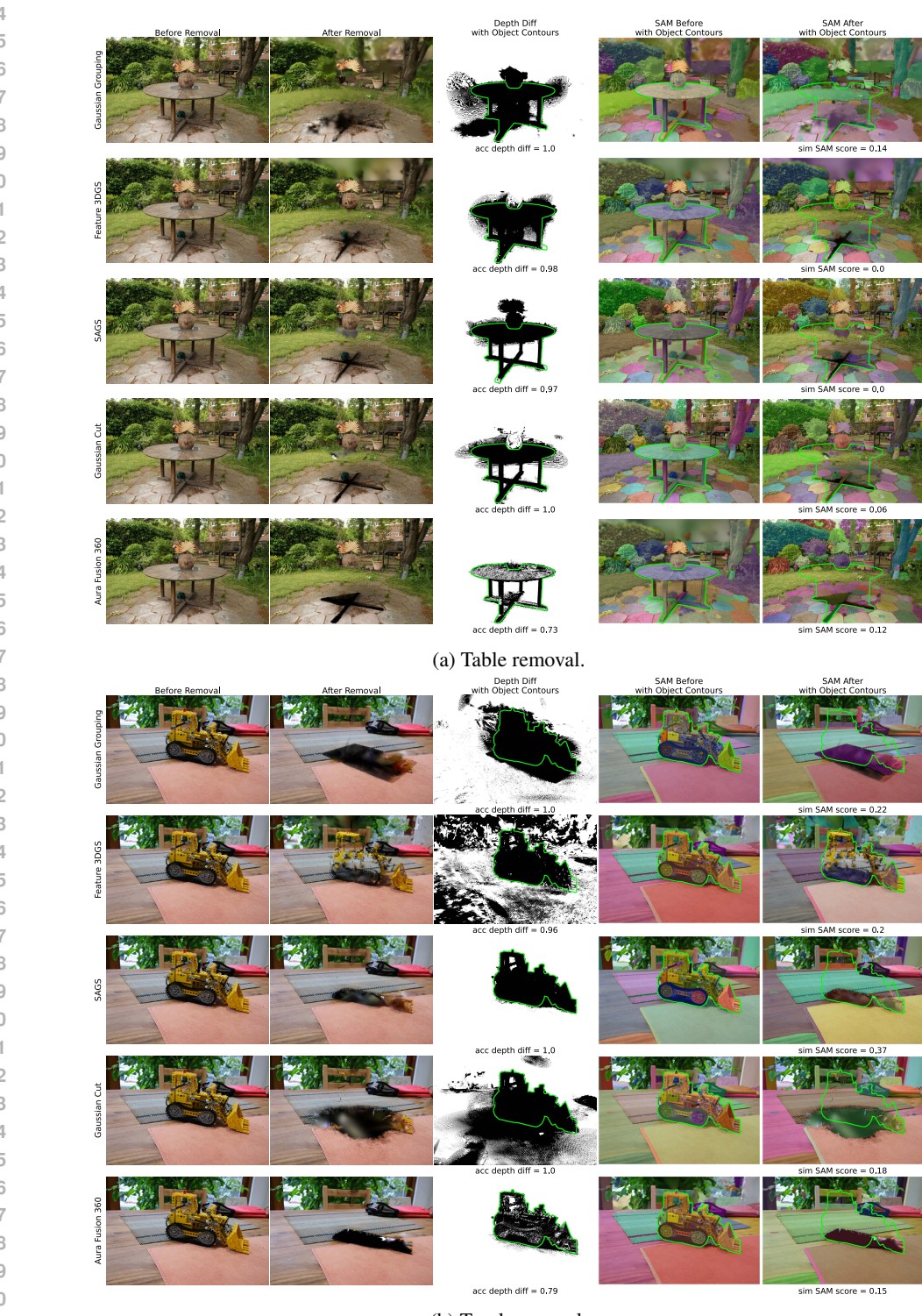

Figure 15: **MipNERF360: Visual comparison of object removal results.** Each row shows outputs from Gaussian Grouping Ye et al. (2024a), Feature 3DGS Zhou et al. (2024), SAGS Hu et al. (2024), Gaussian Cut (GC) Jain et al. (2024), and Aura Fusion Wu et al. (2025). Triplets include the before removal render, removal result, and evaluation via depth difference accuracy and SAM Kirillov et al. (2023) similarity. GC Landrieu & Obozinski (2017) performs best overall, though results across methods are comparable.