# OpenReview forum: "Remove360: Benchmarking Residuals After Object Removal in 3D Gaussian Splatting"
_ICLR.cc/2026/Conference — ICLR 2026 Conference Withdrawn Submission_

### Official Review · Reviewer_8dHo · 2025-10-25

**Soundness:** 3
**Presentation:** 3
**Contribution:** 3
**Rating:** 6
**Confidence:** 3

**Summary:**

The paper introduces a new benchmark and evaluation framework for assessing the effectiveness of object removal techniques in 3D Gaussian Splatting, focusing on whether semantic residuals remain after the removal process. It presents Remove360, a dataset of real-world indoor and outdoor scenes with pre- and post-removal RGB images and object masks.  The paper evaluates several existing object removal methods using a combination of semantic, segmentation, and depth-based metrics to quantify the presence of unintended semantic traces after object removal. The findings highlight limitations in current 3D object removal techniques and emphasize the need for more robust solutions.

**Strengths:**

-The paper addresses a critical and previously underexplored aspect of 3D scene editing – the presence of semantic residuals after object removal, particularly relevant for privacy-preserving applications.
-The proposed evaluation framework combines multiple complementary metrics (semantic, segmentation, and depth-based) to provide a thorough assessment of object removal effectiveness.
-The Remove360 dataset fills a gap by providing a diverse set of real-world indoor and outdoor scenes with pre/post-removal captures and object masks, better reflecting real-world complexities than existing datasets.

**Weaknesses:**

-The evaluation metrics rely on off-the-shelf semantic segmentation models, which may introduce errors and limit the robustness of the evaluation. The paper acknowledges this but could benefit from further discussion of potential biases and limitations.

-Limited Analysis of Failure Cases: While the paper mentions failure cases (residual artifacts, incomplete removal, over-smoothing), a more detailed analysis of specific failure modes would be valuable.

**Questions:**

-How sensitive are the evaluation metrics to errors in the ground-truth object masks or the predictions of the off-the-shelf semantic segmentation models? Can you provide quantitative results on this sensitivity?

-The paper mentions combining metrics but doesn't provide details on how this is done. Are the metrics equally weighted, or is there a more sophisticated combination strategy? What is the justification for the chosen combination method?

-Can you provide a more detailed analysis of specific failure cases observed in the experiments, including potential reasons for these failures?

---

### Official Review · Reviewer_CQoP · 2025-10-30

**Soundness:** 2
**Presentation:** 3
**Contribution:** 2
**Rating:** 4
**Confidence:** 5

**Summary:**

This paper investigates the persistence of semantic information after object removal in 3D scene reconstruction, with a particular focus on privacy-preserving 3D Gaussian Splatting. The authors introduce Remove360, a new benchmark dataset containing real-world indoor and outdoor scenes with paired pre- and post-removal RGB images and object masks. In addition, they propose a set of four quantitative metrics based on semantic segmentation drop, recognition accuracy, instance mask similarity, and depth change to evaluate how much semantic information remains after object removal. Using five state-of-the-art 3DGS-based methods, the authors show that all existing approaches leave detectable semantic residuals, even when removed objects are visually absent. The work highlights key limitations in current 3D editing pipelines and provides a valuable benchmark for developing future privacy-aware scene manipulation methods.

**Strengths:**

1. A new dataset is released to measure the object removal, containing indoor and outdoor scene, which boosts the community.
2. This paper is well-written and the measurement method is easy to understand.

**Weaknesses:**

1. In the introduction, the definition of semantic residuals is unclear and lacks a rigorous mathematical or task-based formulation, which is critical for understanding and designing the subsequent evaluation metrics.

2. Although semantic residuals in 3DGS have been visualized, a deeper analysis is required to understand their underlying causes.

3. In the related work section, the authors only discuss 3D reconstruction and semantics in privacy protection applications. Object removal, however, is a subcategory of 3D scene editing, such as NeRF-based or 3DGS-based editing, and the relationships between these relevant studies and the current work should be discussed in more depth.

4. The authors mention privacy challenges across different domains, but a systematic discussion of how existing methods protect privacy is missing.

5. In the metric definition, the authors propose using semantic object recognition to measure the mIoU drop for identifying semantic residuals. However, segmentation performance is highly dependent on the chosen model, and altering the segmentation model could lead to inconsistent score drops, thereby introducing bias into the proposed metric.

6. Some important statistics, such as the number of objects in the dataset, are not provided.

7. While the experiments are comprehensive, they are entirely built upon existing techniques, with no newly designed method to demonstrate the authors’ own contribution.

8. The ablation studies are insufficient. In the metric definition, the authors employ several hyperparameters, yet key ones, such as the depth threshold, are not analyzed, which raises concerns about the robustness of the proposed metric.

9. The paper states that “operating on the 3D model offers privacy at a reasonable computational cost and better reconstruction quality.” However, a quantitative comparison of computational costs between the proposed 3D method and image-based editing is not provided.

10. In the discussion section, the authors mention that semantic residuals are difficult to eliminate, but no potential solutions or future directions are suggested.

**Questions:**

1. The definition of semantic residuals is not clear.
2. This work only compares among existing methods, no technique novelty.
3. The introduction of the proposed method are incomplete .
4. This work requires more ablation studies.

---

### Official Review · Reviewer_NHx4 · 2025-10-31

**Soundness:** 2
**Presentation:** 2
**Contribution:** 2
**Rating:** 4
**Confidence:** 3

**Summary:**

This paper aims to quantify the semantic traces remaining after object removal in 3D Gaussian Splatting. The core argument is that even when an object is visually removed successfully, invisible "semantic residuals" may still persist in the scene data. To address this challenge, the research introduces an evaluation framework and releases the Remove360 dataset, which contains pre-/post-removal RGB images and object-level masks from real-world scenes.

**Strengths:**

1. The paper focuses on the critical issue of invisible "semantic residuals" that may persist in scene data after 3D removal. To this end, it proposes a comprehensive evaluation from multiple dimensions, including semantics and segmentation.
2. The paper releases the Remove360 dataset, which includes pre-/post-removal RGB images and object-level masks from real-world scenes. Providing these scenes after physical object removal to serve as ground truth offers a solid basis for evaluating removal capabilities. It is also noteworthy that the dataset has garnered significant attention, with over 2,600 downloads in the last month.

**Weaknesses:**

1. The paper lacks a comparison with previous evaluation metrics such as PSNR, SSIM, and LPIPS. It would strengthen the paper's argument to demonstrate cases where the difference lies between these traditional metrics and the proposed metrics.
2. Regarding the paper's assertion that "the model can detect residual information invisible to the human eye": It is possible that the inpainted shape is similar to the original, or that artifacts introduced during the removal and inpainting process coincidentally trigger erroneous recognition by the segmentation model. These issues may stem from errors in the segmentation model itself, rather than flaws in the removal algorithm.
3. Concerning post-removal recognizability: If the removed area were replaced entirely with noise, it would also be unrecognizable. In such a scenario, the reliability of metrics like IoU_drop and sim_SAM would be questionable, as they might not distinguish a successful, context-aware inpainting from simple noise filling.

**Questions:**

My main questions concern the reliability of IoU_drop and sim_SAM, and the comparison with previous pixel-level metrics like PSNR, SSIM, and LPIPS. Please refer to the weaknesses for details.

---

### Official Review · Reviewer_mkkv · 2025-11-03

**Soundness:** 2
**Presentation:** 3
**Contribution:** 3
**Rating:** 4
**Confidence:** 4

**Summary:**

This paper introduces a new dataset and a suite of metrics for evaluating the quality of object removal in 3D Gaussian Splatting. It emphasizes the importance of achieving high-quality, privacy-preserving object removal by examining whether a removed object is truly eliminated or leaves behind semantic residuals that reveal its presence. The proposed metrics jointly assess both low-level visual signals (depth) and high-level semantic consistency, providing a holistic evaluation of removal fidelity. The newly released Remove360 dataset, comprising 11 diverse indoor and outdoor scenes, serves as a comprehensive benchmark for assessing and comparing existing 3DGS-based object removal methods.

**Strengths:**

- Remove360

The paper comes with a real world dataset for object removal, i.e., Remove360. The dataset provides a diverse collection of both indoor and outdoor scenes under different lightings with objects pre- and post- removal. It also comes with GT annotations of objects being removed. Remove360 focuses on the multi-object removal scenario which is a better match for its everyday usage. The dataset could serve as a baseline benchmarking dataset for future works.

- Low and high level metrics

The paper proposes several metrics for object removal evaluation. These metrics cover a wide range of quality assessments, from low level depth consistency to high level semantic labelings. Specifically, it evaluates the segmentation accuracy, semantic accuracy, and depth accuracy. Together, it provides a comprehensive, multi-level evaluation framework for quantifying the quality of 3D object removal.

**Weaknesses:**

- Regions Outside the Object

A common issue in object removal is that shadows, reflections, or indirect effects of the removed objects often remain in the scene. These residual pixels lie outside the object’s mask, yet can still reveal visual or semantic cues about what was removed. The paper’s proposed metrics primarily focus on evaluating changes within the masked region, without explicitly considering these contextual regions outside the mask. Such unaddressed residuals may still compromise privacy or expose identifiable traces. For instance, as shown in Figure 2, leftover shadows remain visible but are not penalized by the current evaluation framework.

- Focus Limited to 3DGS-Based Methods

The paper’s evaluation framework and experiments concentrate exclusively on 3DGS–based object removal techniques. While this focus provides depth and clarity, it overlooks alternative paradigms such as 3D inpainting or neural field completion methods, which also play an important role in 3D scene editing and privacy-preserving reconstruction. 3D inpainting method, e.g., Inpaint3D, takes advantage of diffusion-based priors to fill in missing regions and handle long-range dependencies such as shadows or reflections. Although 3D consistency is not guaranteed, these two types of methods are on the different ends of the spectrum and should both be evaluated.

**Questions:**

Overall the paper comes with a nice collection of metrics and dataset for 3D object removal benchmarking. I have some questions to be answered:

- How do the authors plan to handle shadows, reflections, and indirect lighting effects that remain outside the object mask after removal?
- Have the authors considered extending their metrics to include contextual regions beyond the mask to better capture these residual cues?
- How does 3D inpainting or neural field completion approaches perform in the benchmark?
- Have the authors conducted any human studies to assess whether semantic residuals are perceptually noticeable in practice?

---

### Note · Authors · 2025-11-12

I have read and agree with the venue's withdrawal policy on behalf of myself and my co-authors.